# Dynamic hydrogen-bonding enables high-performance and mechanically robust organic solar cells processed with non-halogenated solvent

Haozhe He[1,2], Xiaojun Li [1,2] ✉, Jingyuan Zhang[1], Zekun Chen[1,2], Yufei Gong [1,2], Hongmei Zhuo[1,2], Xiangxi Wu[1,2], Yuechen Li [1,3], Shijie Wang[4], Zhaozhao Bi [4], Bohao Song[5], Kangkang Zhou[6], Tongling Liang[2,7], Wei Ma[4], Guanghao Lu [5], Long Ye [6], Lei Meng [1,2], Ben Zhang[8], Yaowen Li [8] & Yongfang Li [1,2,8] ✉

Developing active-layer systems with both high performance and mechanical robustness is a crucial step towards achieving future commercialization of flexible and stretchable organic solar cells (OSCs). Herein, we design and synthesize a series of acceptors BTA-C6, BTA-E3, BTA-E6, and BTA-E9, featuring the side chains of hexyl, and 3, 6, and 9 carbon-chain with ethyl ester end groups respectively. Benefiting from suitable phase separation and vertical phase distribution, the PM6:BTA-E3-based OSCs processed by *o*-xylene exhibit lower energy loss and improved charge transport characteristic and achieve a power conversion efficiency of 19.92% (certified 19.57%), which stands as the highest recorded value in binary OSCs processed by green solvents. Moreover, due to the additional hydrogen-bonding provided by ethyl ester side chain, the PM6:BTA-E3-based active-layer systems achieve enhanced stretchability and thermal stability. Our work reveals the significance of dynamic hydrogen-bonding in improving the photovoltaic performance, mechanical robustness, and morphological stability of OSCs.

Organic solar cells (OSCs) have garnered widespread concern as a promising power generation technology[1–8]. The biggest advantages of OSC to complement inorganic photovoltaics in the future lie in the intrinsic flexibility of organic active-layer materials, which not only enables the roll-to-roll processing and low-temperature manufacturing but also allows the integration of OSC devices in various forms inaccessible to inorganic devices. These applications mainly include extremely flexible and stretchable devices like wearable energy source, portable electronics, agricultural greenhouses, and biomedical applications[9,10]. In recent years, thanks to the ground-breaking progress in materials design and device engineering, the power conversion efficiency (PCE) of OSCs has experienced a remarkable surge and exceeded 19%[11–21]. For future commercialization and leveraging the advantages of OSCs, it is crucial to achieve stretchable active-layer

[1]CAS Key Laboratory of Organic Solids, Institute of Chemistry, Chinese Academy of Sciences, Beijing, China. [2]School of Chemical Science, University of Chinese Academy of Sciences, Beijing, China. [3]School of Materials Science and Engineering, Shaanxi Normal University, Xi'an, China. [4]State Key Laboratory for Mechanical Behavior of Materials, Xi'an Jiaotong University, Xi'an, China. [5]Frontier Institute of Science and Technology, and State Key Laboratory of Electrical Insulation and Power Equipment, Xi'an Jiaotong University, Xi'an, China. [6]School of Materials Science and Engineering, Tianjin Key Laboratory of Molecular Optoelectronic Sciences, Tianjin University, Tianjin, China. [7]Center for Physicochemical Analysis and Measurement, Institute of Chemistry, Chinese Academy of Sciences, Beijing, China. [8]Laboratory of Advanced Optoelectronic Materials, College of Chemistry, Chemical Engineering and Materials Science, Soochow University, Suzhou, China. ✉e-mail: lixiaojun@iccas.ac.cn; liyf@iccas.ac.cn

systems with both high efficiency and mechanical robustness. However, the active-layer system of the state-of-the-art OSCs with high performance can hardly meet the demands of flexible electronics, such as the typical polymer donor and small molecule acceptor (SMA) systems usually suffer from low stretchability (i.e., crack onset strain (COS) < 4%)[22,23].

Recently, there has been a growing awareness of the importance of mechanical robustness, and a variety of investigations have been reported aimed at mechanically reliable organic active-layers[24–26]. One of the strategies is the addition of insulating polymers (like PDMS, PAE etc.) as the third component into the active-layer to improve mechanical robustness[27–29]. These flexible insulating polymer chains can enhance molecular chain entanglement in the amorphous region, and facilitate polymer chain movement under external strain, which is beneficial for improving the ductility of active-layer. However, when the proportion of added insulating polymers exceeds a certain range (typically 10%), the efficiency of the OSCs will significantly drop, therefore making it challenging to strike a balance between device performance and mechanical stability[30–33]. Structurally, organic semiconductors generally feature an exquisitely rigid conjugated backbone, which is detrimental to obtaining the high stretchability of the materials. Consequently, another reported strategy is to incorporate flexible non-conjugated segments into the strongly conjugated backbones of polymer donors for high-performance and mechanically robust OSCs[34]. However, this strategy dramatically affects the π-conjugation and engenders unfavorable π-π stacking, which will sacrifice the charge transport capability of materials[35]. Additionally, the introduction of non-conjugated segments also limits the reproducible synthesis. To address these issues, dynamic chemical bonds incorporating flexible side chains should be considered and introduced into the design and synthesis of organic semiconductors. Under the action of external forces, the dynamic bonds can readily be broken to allow energy dissipation upon strain, making the system more tolerant to mechanical stimuli[36]. This method does not dramatically affect the π-conjugation compared with the insertion of conjugation-breaking units into the conjugated backbone and will be possible to achieve both excellent mechanical compliance and high charge mobility.

Hydrogen-bonding (H-bonding), as a type of dynamic chemical bond, is a dynamic, directional, and specific non-covalent bond. The energy of H-bonding (approximately $10-65\,\text{kJ mol}^{-1}$) is stronger than that of most other non-covalent bonds such as the π-π interaction (typically below $10\,\text{kJ mol}^{-1}$) and Van der Waals' force (typically below $8\,\text{kJ mol}^{-1}$)[37]. The high energy and the directionality of H-bonding would affect the molecular conformation and stacking, which are crucial for active-layer morphology control and charge transport dynamics in the OSCs. Moreover, in comparison to covalent bonds, H-bonding is thermodynamically reversible and there is a spontaneously dynamic equilibrium of association/disassociation within H-bonding, showcasing its inherent flexibility and self-assembling capabilities. When subjected to mechanical strain or stimuli, the dynamic H-bonding can effortlessly be disrupted, enabling efficient dissipation of energy under strain and hence bestowing materials with excellent mechanical compliance. On one hand, H-bonding could modify the intermolecular interaction through multiple interactions, thereby manipulating the stacking behaviors of molecules and facilitating charge carrier transport. On the other hand, the resultant non-covalent intermolecular crosslinking network could counteract the external stress-induced damage to some extent and enhance the morphological stability and mechanical robustness within the systems[23,38]. Recently, Thompson et al. reported a H-bonding modification strategy of introducing the thymine side chain terminated 6,7-difluoro-quinoxaline (Q-Thy) third component into polymer donor PM7 (Poly[(2,6-(4,8-bis(5-(2-ethylhexyl-3- chloro) thiophen-2-yl)-benzo[1,2-b:4,5-b]dithiophene))-alt-(5,5-(1,3-di-2-thienyl-5,7-bis(2-ethylhexyl)benzo[1,2-c:4,5-c]dithiophene-4,8-dione)]), and the

beneficial effects of H-bonding on the stretchability of the active-layer were confirmed[22]. However, the random ternary copolymerization would exacerbate the polymer batch-to-batch variation with molecular weight difference or molecular structure defect and present challenges for the repeatability of synthesis as well as the reliability of device photovoltaic performance and mechanical robustness[39–43].

Differing from the polymer materials, the SMAs possess a definite molecular structure which allows them to have good repeatability and to more effectively introduce H-bonding into the active-layer of OSCs. However, there was limited research on H-bonding modification for SMAs, since they typically have strong π-π intermolecular interaction and the introduction of H-bonding into the SMAs will further strengthen intermolecular interaction, which poses a huge challenge to coordinate the crystallinity of SMAs and corresponding blend film morphology. In this work, we introduce dynamic chemical bonds into the flexible side chain of the benzotriazole (BTA) central core of A-DA'D-A type SMAs, by selecting the ethyl ester group as the unit providing H-bonding interactions, combining with the flexible alkyl chains. We design and synthesize a series of acceptors, namely BTA-C6 with hexyl chain, and BTA-E3, BTA-E6, and BTA-E9 with different lengths of ethyl ester side chains, to explore the influence of dynamic H-bonding from the ethyl ester side chains and the flexible side chain length on the photovoltaic properties and mechanical stability of the OSCs. As depicted in Fig. 1a, the *N*-substituted side chains of the BTA central core for the SMAs of BTA-E3, BTA-E6, and BTA-E9 are ethyl propanoate, ethyl hexanoate, and ethyl nonanoate respectively, while BTA-C6 with an *n*-hexyl side chain on the BTA central core was synthesized as control SMA. Among the three SMAs of BTA-E3, BTA-E6, and BTA-E9 with ethyl ester side chains, the ethyl propanoate side chain of BTA-E3 possesses the same length with the *n*-hexyl side chain of BTA-C6. In comparison with BTA-C6, a gradual blueshift both in solution and film absorption spectra was observed from BTA-E9 to BTA-E6 and then to BTA-E3, while the energy levels of each acceptor were measured to be almost identical. Then, in order to compare the photovoltaic performance of the four acceptors, these SMAs-based OSCs with PM6 as polymer donors were fabricated using the non-halogenated solvent *o*-xylene as processing solvent. In comparison with the BTA-C6-based OSC, the device based on BTA-E3 with the same side chain length as BTA-C6 demonstrated more efficient exciton disassociation and charge carrier transport as well as more favorable blend film morphology, leading to a remarkable PCE of 19.92% by employing 2PACz as the hole transport layer. The results indicate that introducing the dynamic H-bonding from the ethyl ester side chain is an effective strategy for promoting the photovoltaic performance of OSCs. While as the side chain length elongating, the BTA-E6, and BTA-E9-based OSCs exhibited gradually higher voltage losses, inadequate charge transport capability, and uneven vertical phase distribution, consequently lead to the gradually inferior photovoltaic performance with the respective PCE of 16.27% and 14.93%, indicating that appropriate length of the ethyl ester side chains is important for the dynamic H-bonding to improve the photovoltaic performance. The PCE of 19.92% (certified value of 19.57%) for the device based on PM6:BTA-E3 is a recorded PCE in binary devices processed with green solvents. Most importantly, the mechanical properties and device stability of the PM6:BTA-E3-based OSCs is significantly enhanced due to the incorporation of ethyl ester side chain, which further validates our strategy of dynamic H-bonding modification as a promising avenue towards achieving high-performance OSCs with outstanding mechanical robustness.

## Results

### Materials synthesis and optoelectronic properties
The molecular structures of BTA-C6, BTA-E3, BTA-E6, and BTA-E9 are illustrated in Fig. 1a, and the synthetic routes and detailed procedures for these acceptors are described in Supplementary Methods and

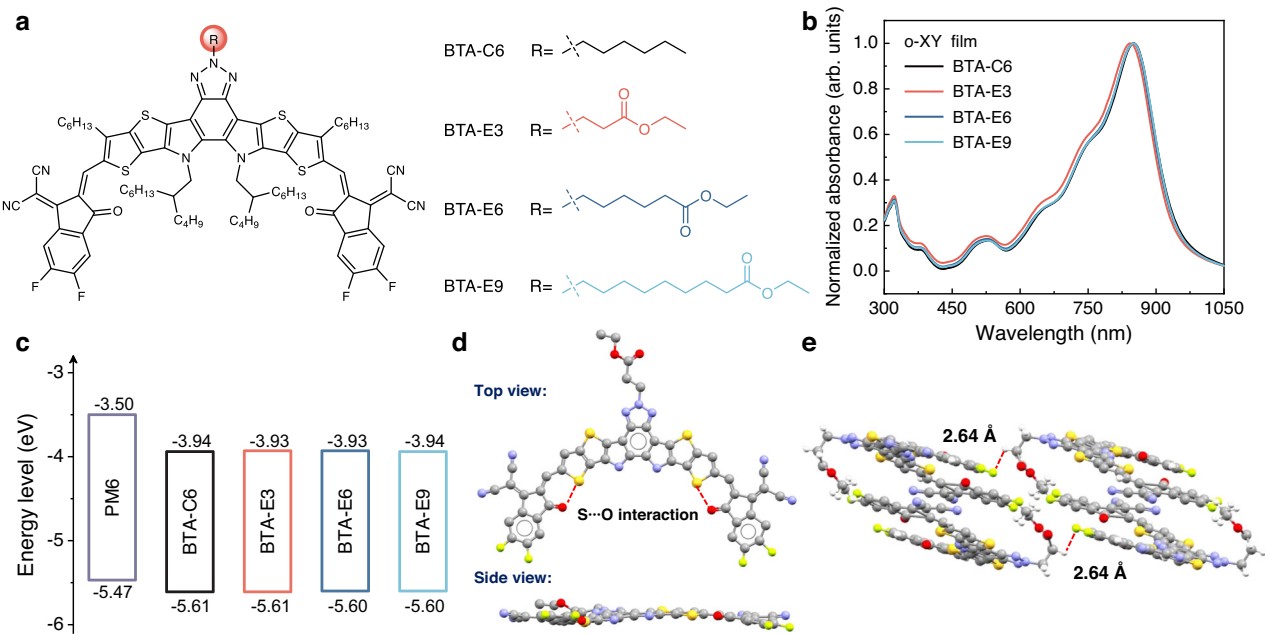

**Fig. 1 | Molecular structures, photophysical properties, and single-crystal structures. a** Molecular structures of BTA-C6, BTA-E3, BTA-E6 and BTA-E9. **b** UV-vis absorption spectra of BTA-C6, BTA-E3, BTA-E6 and BTA-E9 films prepared from *o*-xylene solutions. **c** Energy level diagrams of PM6, BTA-C6, BTA-E3, BTA-E6 and BTA-E9. **d** The mono-molecular structure of BTA-E3 in top and side view. **e** The single-crystal stacking diagrams in T/b stacking mode of BTA-E3. Source data are provided as a Source Data file.

**Table 1 | Summary of optical properties and electronic energy levels of BTA-C6, BTA-E3, BTA-E6 and BTA-E9**

| Acceptor | *o*-xylene solutions | | *o*-xylene films | | $E_g^{opt}$ (eV) | $E_{HOMO}$ (eV) | $E_{LOMO}$ (eV) |
|---|---|---|---|---|---|---|---|
| | $\lambda_{max}$ (nm) | $\lambda_{onset}$ (nm) | $\lambda_{max}$ (nm) | $\lambda_{onset}$ (nm) | | | |
| BTA-C6 | 736 | 777 | 852 | 947 | 1.309 | −5.61 | −3.94 |
| BTA-E3 | 733 | 774 | 843 | 939 | 1.321 | −5.61 | −3.93 |
| BTA-E6 | 735 | 776 | 849 | 944 | 1.314 | −5.60 | −3.93 |
| BTA-E9 | 736 | 777 | 850 | 945 | 1.312 | −5.60 | −3.94 |

Supplementary Figs. 1–3. The molecular structures of all the SMAs were confirmed by nuclear magnetic resonance (NMR) spectroscopy and matrix-assisted laser desorption ionization time of flight mass spectrometry (MALDI-TOF MS), and the corresponding ¹H NMR spectra, ¹³C NMR spectra and mass spectra are shown in Supplementary Figs. 4–35. The four SMAs exhibit good solubility in non-halogenated solvent *o*-xylene, ensuring the processability of corresponding OSCs. Then the thermogravimetric analysis (TGA) revealed the thermal stability of these SMAs, and the results are shown in Supplementary Fig. 36. The thermal decomposition temperatures with a 5% weight loss of BTA-C6, BTA-E3, BTA-E6, and BTA-E9 were measured to be 329.7 °C, 325.6 °C, 321.5 °C and 321.1 °C respectively, ensuring sufficient thermal stability as photovoltaic materials in OSCs.

The UV-vis absorption spectra of the dilute solutions in *o*-xylene and films of the four SMAs are presented in Supplementary Fig. 37 and Fig. 1b respectively, and the detailed data of the optical properties are listed in Table 1. Specifically, the absorption profile of BTA-E9 in *o*-xylene dilute solution was found to be identical to that of BTA-C6, while BTA-E6 and BTA-E3 exhibited a gradual and slight blueshift in their solution absorption spectra. The observed results may be ascribed to the solvation effect and electron-withdrawing effect of the ethyl ester group[44,45]. Furthermore, in the absorption spectra of the films prepared from their *o*-xylene solutions, the film absorptions of all the SMAs exhibited a significant bathochromic shift of ca. 110 nm compared to their solution absorption, indicating the strong molecular

aggregation in the solid state resulting from their intermolecular interactions. Additionally, the consistent blueshift trend in film absorption was observed from BTA-E9 to BTA-E6 and then to BTA-E3. All the neat films displayed a strong and broad absorption spanning from 600 to 950 nm, and the maximum film absorption peaks for BTA-C6, BTA-E3, BTA-E6, and BTA-E9 located at 852 nm, 843 nm, 849 nm, and 850 nm, respectively. The optical bandgap ($E_g^{opt}$) of BTA-E3 (1.321 eV), calculated by $E_g^{opt} = 1240/\lambda_{onset}$, is slightly larger than that of BTA-C6 (1.309 eV), BTA-E6 (1.314 eV), and BTA-E9 (1.312 eV).

The highest occupied molecular orbital (HOMO) and lowest unoccupied molecular orbital (LUMO) energy levels ($E_{HOMO}/E_{LUMO}$) of the photovoltaic materials were measured by electrochemical cyclic voltammetry[46], and their cyclic voltammograms were shown in Supplementary Fig. 38. From the onset oxidation/reduction potentials shown in the cyclic voltammograms, the $E_{HOMO}/E_{LUMO}$ values were calculated to be −5.47 eV/−3.50 eV for PM6; −5.61 eV/-3.94 eV for BTA-C6; −5.61 eV/−3.93 eV for BTA-E3; −5.60 eV/−3.93 eV for BTA-E6; and −5.60 eV/−3.94 eV for BTA-E9 (see Table 1). The energy level diagrams of PM6 and the SMAs are illustrated in Fig. 1c. The BTA-series acceptors demonstrated a nearly consistent alignment of HOMO and LUMO energy levels and exhibited excellent compatibility with PM6, which implies that the substitution of side chains on BTA central core has negligible impact on their frontier molecular orbital energy levels.

In order to investigate the H-bonding influence on molecular stacking behavior, we obtained the single-crystals of BTA-E3 (CCDC number: 2333735) by slowly diffusing isopropanol into BTA-E3

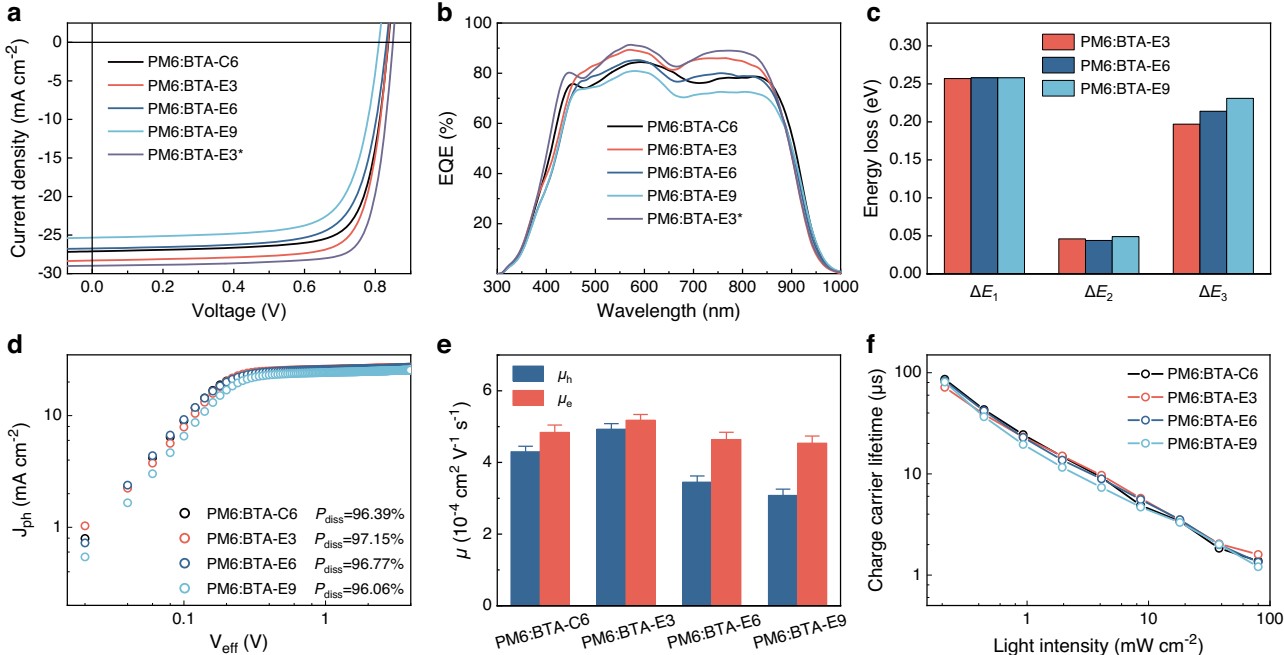

**Fig. 2 | Device photovoltaic performance, energy losses analysis, and charge dynamics. a** *J-V* curves of the optimal OSCs under the illumination of AM 1.5 G, 100 mW cm⁻². **b** EQE curves of the optimal OSCs (where the symbol * represents the hole-transfer layer is 2PACz). **c** Statistical diagram of energy loss. **d** $J_{ph}$ versus $V_{eff}$ curves of the corresponding OSCs. **e** Hole and electron mobilities of the corresponding OSCs. Error bars represent the standard error of the mean (*n* = 5). **f** Charge carrier lifetime curves under different light intensities. Source data are provided as a Source Data file.

chloroform solution (Supplementary Fig. 39). The chemical structure of BTA-E3 was confirmed through single-crystal X-ray diffraction (XRD) analysis. As described in Fig. 1d, BTA-E3 exhibits the crescent-shaped planar molecular geometry with the intramolecular S···O non-covalent interactions. Impressively, as shown in Supplementary Fig. 40, the ethyl ester side chain on the BTA central core exhibits two different orientations: one conformation involves an extension of the ethyl ester side chain along the conjugated plane, while another conformation features a protrusion of the ethyl ester side chain outwards from the conjugated plane. Moreover, BTA-E3 exhibits two types π-π stacking modes, including the terminal/terminal group (T/T) interactions as well as the terminal group and thieno[3,2-b]thiophene unit (T/b) stacking. The corresponding π-π stacking distances were measured to be 4.21 Å for T/T mode and 3.15 Å for T/b mode, respectively. In T/b stacking mode, a ringlike dimer conformation is formed by two molecules through non-covalent interaction between the terminal group and ethyl ester side chain (Fig. 1e). Furthermore, the C-H···F non-covalent H-bonding interaction with a distance of 2.64 Å is generated between the F atom of the terminal group and the H atom of methylene adjacent to the carbonyl group in another dimer. Note that there is no such H-bonding interaction has been ever found and reported between the alkyl side chain on the BTA central core and terminal group in the BTA-based acceptors[47,48]. These results suggest that the introduction of ethyl ester side chain into the central core of BTA-E3 could provide additional intense intermolecular non-covalent H-bonding interactions, which contributes to the dense and ordered three-dimensional molecular stacking network, thereby facilitating efficient charge transport. In addition, when blended with polymer donor, this type of additional H-bonding of BTA-E3 may cause its blend film to form a non-covalent cross-linked network, thereby improving the film stability[49].

## Photovoltaic performance and charge dynamics

To investigate the effect of intermolecular H-bonding and flexible side chain length on the photovoltaic performance of OSCs, we fabricated the OSCs with a conventional structure of indium tin oxide (ITO)/poly(3,4-ethylenedioxythiophene): poly(styrenesulfonate) (PEDOT:PSS) or [2-(9-H-carbazol-9-yl)ethyl] phosphonic acid (2PACz)/ PM6:BTA-series acceptors/N,N'-Bis{3-[3-(Dimethylamino) propylamino]propyl}perylene-3,4,9,10-tetracarboxylic diimide (PDINN)/Ag. PM6 was selected as polymer donor due to its complementary absorption and well-matched energy levels with all the SMAs. It needs to be emphasized that the environmentally friendly green solvent *o*-xylene was employed as the processing solvent (Supplementary Notes I and II) and 1,8-diiodooctane (DIO) was added as an additive to optimize the photovoltaic performance of the OSCs. The current density-voltage curves (*J-V*) of the optimized OSCs are depicted in Fig. 2a and the corresponding photovoltaic performance parameters are listed in Table 2. The PM6:BTA-C6-based devices exhibited a PCE of 17.29% with an open-circuit voltage ($V_{oc}$) of 0.837 V, a short-circuit current density ($J_{sc}$) of 27.12 mA cm⁻² and a filling factor (FF) of 76.17%. Notably, replacing the *n*-hexyl side chain with the same length ethyl propanoate side chain, the PM6:BTA-E3-based devices demonstrated enhanced photovoltaic performance with the $V_{oc}$ of 0.837 V, $J_{sc}$ of 28.31 mA cm⁻², FF of 77.48% and ultimately achieved a PCE of 18.36%. While as the ethyl ester side chain length elongated, the OSCs based on PM6:BTA-E6 and PM6:BTA-E9 exhibited gradually inferior photovoltaic performance with the respective PCE of 16.27% and 14.93%. The remarkable PCE of PM6:BTA-E3-based OSCs was a result of the collective contribution from all the photovoltaic performance parameters, in which the higher $V_{oc}$ primarily results from the reduced voltage loss, and the better FF and $J_{sc}$ should be ascribed to the enhanced charge extraction and transport as well as the more favorable donor-acceptor phase separation and vertical phase distribution. These aspects will be further elaborated in subsequent sections for a comprehensive understanding. Impressively, with further optimization by substituting PEDOT: PSS with 2PACz as the hole transport layer, the PM6:BTA-E3-based devices achieved an unexpectedly excellent PCE of 19.92% with a $V_{oc}$ of 0.852 V, a $J_{sc}$ of 28.99 mA cm⁻² and an FF of 80.63%. This PCE stands as the highest recorded PCE in the devices

**Table 2 | Photovoltaic parameters of the optimized OSCs under the illumination of AM 1.5 G, 100 mW cm⁻²**

| Active Layer | $V_{oc}$ (V) | $J_{sc}$ (mA cm⁻²) | $J_{sc}$[a] (mA cm⁻²) | FF (%) | PCE[b] (%) |
|---|---|---|---|---|---|
| PM6:BTA-C6 | 0.837 (0.835 ± 0.003) | 27.12 (26.82 ± 0.19) | 26.05 | 76.17 (76.09 ± 0.37) | 17.29 (17.04 ± 0.13) |
| PM6:BTA-E3 | 0.837 (0.836 ± 0.002) | 28.31 (28.06 ± 0.22) | 27.12 | 77.48 (76.98 ± 0.26) | 18.36 (18.06 ± 0.13) |
| PM6:BTA-E6 | 0.832 (0.832 ± 0.002) | 26.74 (26.56 ± 0.13) | 25.77 | 73.15 (72.93 ± 0.26) | 16.27 (16.11 ± 0.13) |
| PM6:BTA-E9 | 0.811 (0.807 ± 0.003) | 25.34 (25.32 ± 0.12) | 24.29 | 72.63 (72.48 ± 0.43) | 14.93 (14.76 ± 0.11) |
| PM6:BTA-E3[c] | 0.852 (0.848 ± 0.002) | 28.99 (28.84 ± 0.15) | 27.92 | 80.63 (80.20 ± 0.24) | 19.92 (19.63 ± 0.11) |
| PM6:BTA-E3[c] | 0.847 | 28.91 | | 79.92 | 19.57 [d] |

[a]Integrated from EQE curves.
[b]Average from 30 devices.
[c]Using 2PACz as hole transport layer.
[d]Certified by Photovoltaic and Wind Power Systems Quality Test Center, IEE, Chinese Academy of Sciences (PWQTC).

processed by green solvents (Supplementary Table 1). The improved PCE of 2PACz-based devices could be attributed to the reduced parasitic absorption of the hole transport layer and the more favorable vertical phase separation of active-layer in devices, which will be discussed later. Furthermore, the optimized PM6:BTA-E3-based device obtained a certified PCE of 19.57% subject to the calibration procedures of Photovoltaic and Wind Power Systems Quality Test Center, IEE, Chinese Academy of Sciences (Supplementary Fig. 41), which stands for the highest recorded PCE in the binary devices processed by green solvents to the best of our knowledge.

Furthermore, the external quantum efficiency (EQE) spectra of these OSCs are shown in Fig. 2b. All the devices exhibited similar broad responses ranging from 350 to 950 nm, but differed in their respective response magnitudes. In particular, the EQE response of the PM6:BTA-E3-based devices significantly surpassed that of other devices, with the EQE values above 80% across the wavelength range of 500–850 nm. The calculated $J_{sc}$ values, by integrating the EQE curves, for the devices based on PM6:BTA-C6, PM6:BTA-E3, PM6:BTA-E6 and PM6:BTA-E9 were 26.05 mA cm⁻², 27.12 mA cm⁻², 25.77 mA cm⁻² and 24.29 mA cm⁻² respectively. By utilizing 2PACz as the hole transport layer, the PM6:BTA-E3-based OSCs demonstrated enhanced EQE responses across all wavelengths with the maximum response value soaring beyond 90%, and achieved an excellent calculated $J_{sc}$ of 27.92 mA cm⁻². Additionally, the calculated $J_{sc}$ values exhibit good concordance with the $J_{sc}$ values measured in $J$-$V$ curves, confirming the credibility of the photovoltaic performance measurement.

To explore the variation of $V_{oc}$ in the OSCs based on PM6:BTA-E3, PM6:BTA-E6, and PM6:BTA-E9, the energy loss ($E_{loss}$) analysis of the corresponding devices was conducted[50]. The total $E_{loss}$ was calculated following the equation $E_{loss} = E_g^{PV} - qV_{oc}^{cal}$, and the corresponding values of PM6:BTA-E3, PM6:BTA-E6, and PM6:BTA-E9 based OSCs were measured to be 0.500 eV, 0.516 eV and 0.538 eV respectively. According to the Shockley-Queisser (S-Q) theory[51], the overall $E_{loss}$ in OSCs can be categorized into three components:$E_{loss} = (E_g^{PV} - qV_{oc}^{SQ}) + (qV_{oc}^{SQ} - qV_{oc}^{rad}) + (qV_{oc}^{rad} - qV_{oc}^{cal}) = \Delta E_1 + \Delta E_2 + \Delta E_3$. Therein, the first part $\Delta E_1$ is inevitable radiative loss above the bandgap and mainly depends on the bandgap of materials and temperature, which can be calculated by the difference between the photovoltaic bandgap ($E_g^{PV}$) and the output voltage in S-Q limit mode ($V_{oc}^{SQ}$). As depicted in Fig. 2c and Supplementary Table 2, $\Delta E_1$ values of the OSCs based on PM6:BTA-E3, PM6:BTA-E6 and PM6:BTA-E9 were determined to be the similar 0.257–0.258 eV. Moreover, the second part $\Delta E_2$ refers to the radiative recombination loss below the bandgap, and the $\Delta E_2$ values were determined as 0.046 eV, 0.044 eV and 0.049 eV for the devices based on PM6:BTA-E3, PM6:BTA-E6 and PM6:BTA-E9. The last component $\Delta E_3$, determined by the electroluminescent external quantum efficiency ($EQE_{EL}$), is the dominating factor among the three components and derives from the non-radiative recombination loss in the devices. The $\Delta E_3$ value can be calculated by $-(kT/q)\ln(EQE_{EL})$, in which $k$ is the Boltzmann constant, $T$ is Kelvin temperature and $EQE_{EL}$

is the electroluminescence external quantum efficiency under dark condition[52]. The $EQE_{EL}$ of the OSCs based on PM6:BTA-E3, PM6:BTA-E6, and PM6:BTA-E9 were estimated to be $5.00 \times 10^{-4}$, $2.49 \times 10^{-4}$ and $1.30 \times 10^{-4}$ (Supplementary Fig. 42) and the corresponding $\Delta E_3$ values were 0.197 eV, 0.214 eV and 0.231 eV respectively. The gradually decreased $EQE_{EL}$ and increased $\Delta E_3$ values with increasing the side chain length of the SMAs suggest that the proper length ethyl ester side chain in BTA-E3 could somehow suppress the non-radiative recombination loss. Consequently, the diminution of non-radiative decay greatly promotes the decrease of $E_{loss}$, and increase of $V_{oc}$ from BTA-E9 to BTA-E6, and then to BTA-E3-based devices.

The dependence of photocurrent density ($J_{ph}$) on the effective voltage ($V_{eff}$) was measured to evaluate the exciton dissociation properties of the OSCs (Fig. 2d)[53]. The exciton dissociation probability ($P_{diss}$) was calculated from the ratio of $J_{ph}$ and the saturated photocurrent density ($J_{sat}$) (Supplementary Table 3), and the $P_{diss}$ value of the OSCs based on PM6:BTA-E3 was estimated to be 97.15%, which is higher than that of the devices based on PM6:BTA-C6 (96.39%), PM6:BTA-E6 (96.77%), and PM6:BTA-E9 (96.06%). Generally, the higher $P_{diss}$ value in OSCs indicates that the exciton could effectively dissociate at the D/A interface, and the most efficient exciton dissociation observed in PM6:BTA-E3-based OSCs could account for their high $J_{sc}$. In addition, the charge extraction characteristics were estimated by transient photocurrent (TPC) measurement. As depicted in Supplementary Fig. 43, the current responses of each device exhibited rapid rise and decay upon exposure to the 100 μs light irradiation pulse. Notably, the PM6:BTA-E3-based OSCs feature the fastest turn-on and turn-off dynamic compared to other devices, suggesting rapid charge generation and extraction processes as well as reduced presence of charge traps within the PM6:BTA-E3-based OSCs. These results imply that the introduction of a suitable length ethyl ester side chain could enhance the exciton dissociation efficiency and improve charge generation and extraction, ensuring the excellent $J_{sc}$ and FF of the OSCs.

Furthermore, the charge carrier mobilities of the optimized blend films were measured using space-charge-limited current (SCLC) method[54]. As illustrated in Fig. 2e and Supplementary Table 4, the average hole ($\mu_h$) and electron ($\mu_e$) mobilities of the PM6:BTA-C6, PM6:BTA-E3, PM6:BTA-E6 and PM6:BTA-E9 blend films were estimated to be $4.30 \times 10^{-4}/4.84 \times 10^{-4}$ cm² V⁻¹ s⁻¹, $4.93 \times 10^{-4}/5.18 \times 10^{-4}$ cm² V⁻¹ s⁻¹, $3.45 \times 10^{-4}/4.64 \times 10^{-4}$ cm² V⁻¹ s⁻¹ and $3.08 \times 10^{-4}/4.54 \times 10^{-4}$ cm² V⁻¹ s⁻¹ respectively. The corresponding blend films exhibited the $\mu_e/\mu_h$ ratios of 1.13, 1.05, 1.35, and 1.47 for each respective blend film. The results showed that all the blend films exhibited relatively high electron mobilities while varied in their hole mobilities. In particular, the PM6:BTA-E3 blend films exhibited higher hole mobilities than the PM6:BTA-C6 blend films counterparts, while the PM6:BTA-E6 and PM6:BTA-E9 blend films possessed gradually reduced hole mobilities, resulting in unmatched electron and hole transport. Consequently, the

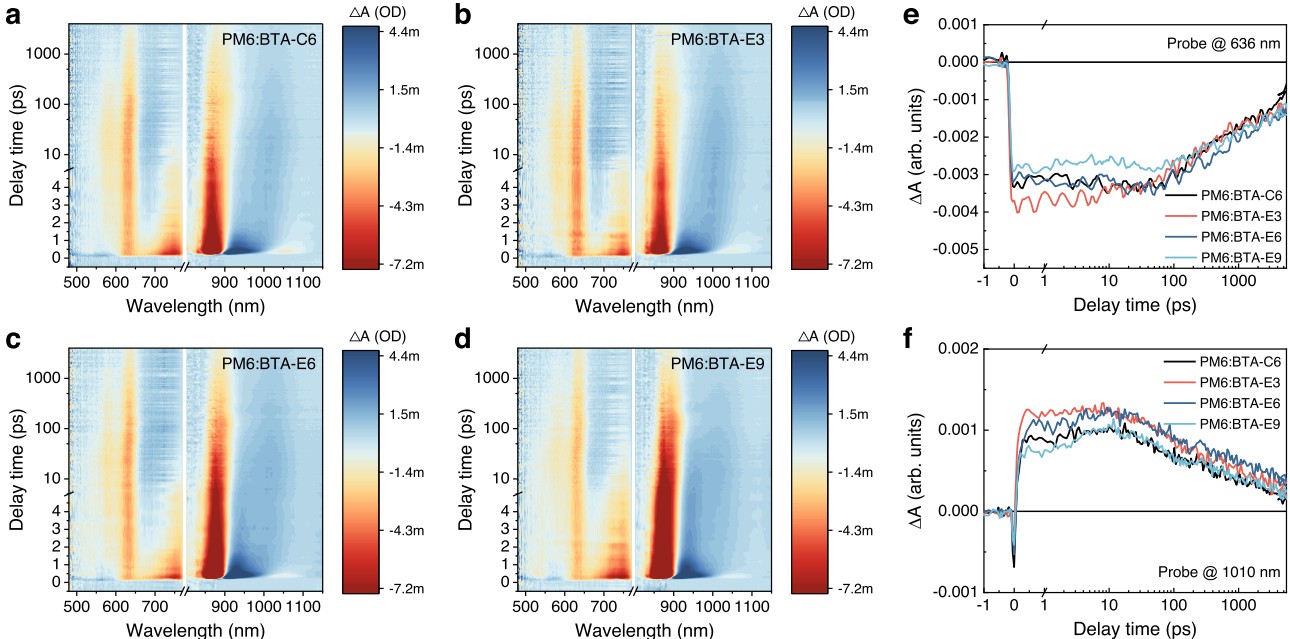

**Fig. 3 | Transient absorption spectroscopy.** 2D transient absorption spectra of the blend films of PM6:BTA-C6 (**a**), PM6:BTA-E3 (**b**), PM6:BTA-E6 (**c**) and PM6:BTA-E9 (**d**). Kinetic traces probing at 636 nm (**e**) and 1010 nm (**f**) for the blend films of PM6:BTA-C6, PM6:BTA-E3, PM6:BTA-E6, and PM6:BTA-E9. Source data are provided as a Source Data file.

lower charge mobilities and unbalanced $\mu_e/\mu_h$ ratios of the PM6:BTA-E6 and PM6:BTA-E9 based OSCs, indicating the inadequate charge transport capability, could potentially lead to the poorer $J_{sc}$ and FF.

Additionally, to investigate the charge carrier properties under operation condition, we measured the photoinduced charge extraction by linearly increasing voltage (photo-CELIV) of the devices[55]. As shown in Supplementary Fig. 44, the extracted charge carrier mobility ($\mu$) was determined as $2.40 \times 10^{-4}$ cm$^2$ V$^{-1}$ s$^{-1}$, $2.50 \times 10^{-4}$ cm$^2$ V$^{-1}$ s$^{-1}$, $2.37 \times 10^{-4}$ cm$^2$ V$^{-1}$ s$^{-1}$ and $2.26 \times 10^{-4}$ cm$^2$ V$^{-1}$ s$^{-1}$ for the OSCs based on PM6:BTA-C6, PM6:BTA-E3, PM6:BTA-E6 and PM6:BTA-E9 respectively. As expected, the observed trend in extracted charge carrier mobility consists with the results obtained from SCLC measurement. The faster and more balanced charge mobilities of the PM6:BTA-E3 and PM6:BTA-C6-based devices are conducive to mitigating charge accumulation and enhancing charge transport, thus could contribute to the improved $J_{sc}$ and FF in their devices.

Moreover, to reveal the charge recombination behavior in the OSCs, the dependence of $J_{sc}$ on irradiated light intensity ($P_{light}$) was measured[56]. The relationship of $J_{sc}$ and $P_{light}$ could be described by the equation: $J_{sc} \propto P_{light}^{\alpha}$, where $\alpha$ is the power law exponent. Supplementary Fig. 45 shows the plots of the logarithm of $J_{sc}$ versus the logarithm of $P_{light}$. For the devices based on PM6:BTA-C6, PM6:BTA-E3, PM6:BTA-E6 and PM6:BTA-E9, the $\alpha$ values were determined to be 0.994, 0.997, 0.992, and 0.989 respectively. In comparison with the PM6:BTA-C6 counterparts, the PM6:BTA-E3-based OSCs exhibited the $\alpha$ value closer to 1, implying the reduced bimolecular recombination within the devices. From the devices based on PM6:BTA-E3 to PM6:BTA-E6 and then to PM6:BTA-E9, a decreasing trend in $\alpha$ was observed as the ethyl ester side chain of the SMAs elongating, which indicates the gradually increased bimolecular recombination within the corresponding OSCs. Besides, the transient photovoltage (TPV) measurement was also employed to evaluate the charge carrier lifetime and charge recombination. Figure 2f illustrates the plots of charge carrier lifetime for the OSCs under varying irradiation intensity. Specifically, the charge carrier lifetimes under the illumination of 100 mW cm$^{-2}$ (1 sun equivalent) were determined as 1.38 μs, 1.60 μs, 1.34 μs, and 1.21 μs for the OSCs based on PM6:BTA-C6, PM6:BTA-E3, PM6:BTA-E6 and PM6:BTA-E9

respectively. Consisting of the dependence of $J_{sc}$ on light intensity measurement, a similar variation trend was observed in the TPV measurement. The PM6:BTA-E3-based devices demonstrated the longest photo-generated carrier lifetime among all the OSCs, indicating the efficient suppression of charge carrier recombination, which is beneficial for obtaining higher $J_{sc}$ and FF in the OSCs.

The femtosecond transient absorption spectroscopy (fsTA) measurement was conducted to explore the charge transfer (CT) behavior and excited state dynamics between the donor and BTA-series acceptors. In particular, each acceptor in the corresponding blend film was selectively photoexcited by the pump light at 810 nm. Figure 3a−d exhibits the 2D fs-TA spectra of the BTA-C6, BTA-E3, BTA-E6, and BTA-E9-based blend films. As depicted in Supplementary Fig. 46, following excitation, the transient absorption spectrum of all the blend films exhibited two distinct ground state bleach (GSB) peaks at 750 nm and 860 nm as well as an excited state absorption (ESA) peak at 920 nm, which belongs to the excited state of acceptors. The peak at 636 nm corresponds to the GSB of the donor PM6, arising from the ultrafast hole transfer from the acceptors to PM6 donor. The GSB peaks at 750 nm and 860 nm and the ESA peak at 920 nm rapidly attenuated in the initial several picoseconds, indicating the occurrence of CT between the donor and acceptors. At later time, the long-live CT predominated the transient absorption spectrum, consisting of a GSB peak at 636 nm and an ESA peak at 700 nm for the donor, as well as a GSB peak at 860 nm and an ESA peak at 1000 nm for the acceptor. Furthermore, Fig. 3e, f presented the kinetic traces of the CT state with the GSB peak for the donor at 636 nm, and the ESA peak for the acceptor at 1010 nm. Upon comparison of the CT state intensity at the 636 nm for donor GSB peak, it is evident that the yield of CT state in the PM6:BTA-E3 blend film surpassed that of the blend films of PM6:BTA-E6 and PM6:BTA-C6, while the PM6:BTA-E9 blend film exhibited a relatively lower CT state yield. The same trend was observed in the kinetic traces at 1010 nm, the CT state intensity of the PM6:BTA-E3 blend film had reached its maximum at the initial period and maintained at maximum for several tens of picoseconds before eventually decaying to the ground state, indicating that the dynamic equilibrium between the generation of CT state and charge recombination had

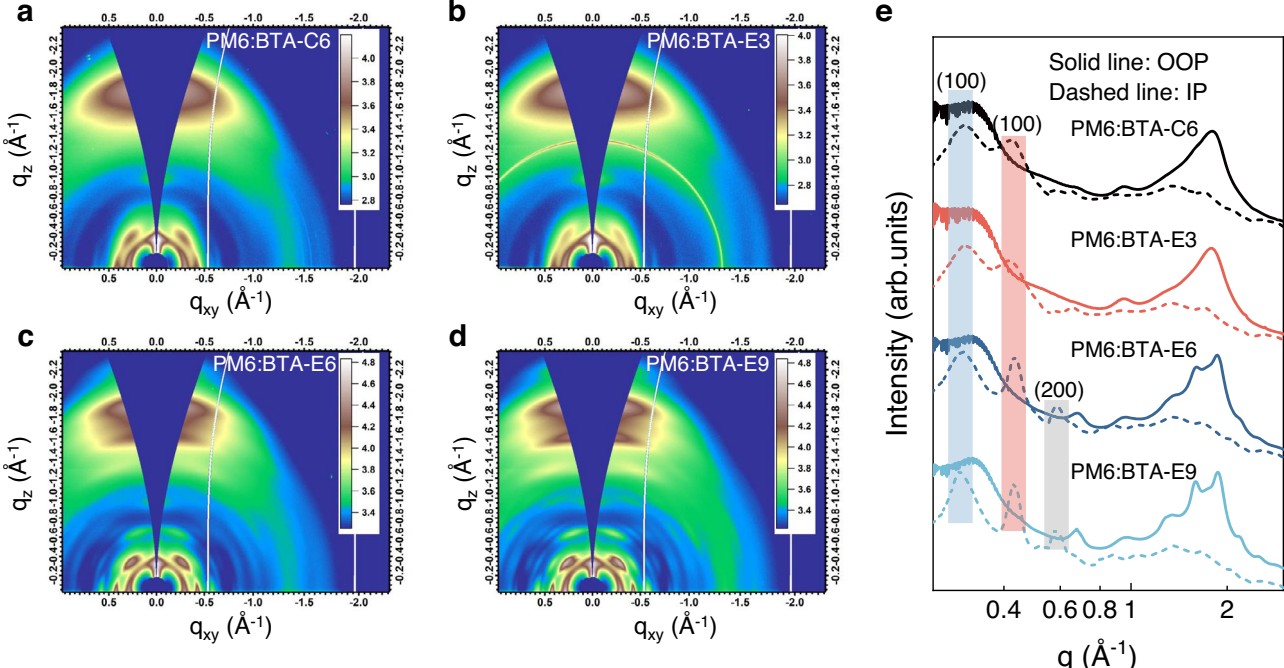

**Fig. 4 | GIWAXS measurements.** 2D GIWAXS patterns of the blend films of PM6:BTA-C6 (**a**), PM6:BTA-E3 (**b**), PM6:BTA-E6 (**c**) and PM6:BTA-E9 (**d**). **e** IP (dashed line) and OOP (solid line) 1D line cut profiles of the 2D GIWAXS data based on the corresponding blend films. The blue, red and gray colored parts represent the (100) peak of lamellar stacking of PM6, the (100) peak of lamellar stacking of acceptors, and the (200) peak of lamellar stacking of PM6 respectively. Source data are provided as a Source Data file.

been reached. In contrast, the rates at which PM6:BTA-E6 and PM6:BTA-C6 blend films reached the maximum intensity of CT state were comparatively slower than that observed in the PM6:BTA-E3 blend film, and the PM6:BTA-E9 blend film exhibited the slowest rate among all the blend films. Based on the observation above, it could be concluded that the distinct side chains could result in the different excited state behavior. The higher CT state yield and prolonged CT lifetime of the PM6:BTA-E3 blend film could facilitate the photogenerated charge carrier extraction and transport, thereby contributing to the higher FF and $J_{sc}$ achieved in the PM6:BTA-E3-based devices.

**Morphology characterization**

In order to explore the relationship underlying the influence of H-bonding interaction and flexible side chain length on molecular crystallinity and film morphology, grazing incidence wide-angle X-ray scattering (GIWAXS) measurements of neat and blend films were performed. As depicted in Supplementary Fig. 47, all the acceptors in neat film exhibited the apparent face-on orientation, with the π-π stacking (010) scatter peaks along the out-of-plane (OOP) direction locating at 1.76 Å⁻¹. The crystalline coherence lengths (CCLs) of π-π stacking were determined using the Scherrer equation and the detailed data were summarized in Supplementary Table 5. The four neat acceptor films exhibited the identical π-π stacking spacing of 3.58 Å, indicating that the side chains on the central core had a negligible influence on the π-π stacking distance. The CCL of π-π stacking in the BTA-E3 neat film was calculated to be 18.72 Å, higher than that of BTA-C6 (17.32 Å), implying that H-bonding interaction of ethyl ester group could facilitate ordered molecular stacking. However, the BTA-E6 and BTA-E9 neat films exhibited weaker ordered π-π molecular stacking, as evidenced by the significantly smaller CCLs of 13.03 Å (BTA-E6) and 13.06 Å (BTA-E9) compared to those of BTA-E3 and BTA-C6 neat films. Moreover, an unexpected peak at 0.45 Å⁻¹ along the OOP direction, potentially ascribing to the (100) peak of lamellar stacking, had been found in both the BTA-E6 and BTA-E9 neat films, signifying the increased priority of edge-on orientation. These observed results

implied that due to their extended ethyl ester side chain positioned distantly from the central conjugated backbone, different packing orientation and stacking modes may arise in BTA-E6 and BTA-E9 neat films.

Furthermore, the GIWAXS measurement of the blend films of the SMAs blended with PM6 was conducted to gain a deep insight of the molecular stacking in their blend films. The two-dimensional (2D) scattering patterns and one-dimensional (1D) line cut profiles are summarized in Fig. 4a–e. All the blend films exhibited the consistent preferential face-on orientation with the neat films according to π-π stacking peak locations. Specifically, the PM6:BTA-E3 blend film demonstrated a higher degree of ordered π-π stacking behavior (with a π-π stacking distance of 3.51 Å and CCL of 20.78 Å) compared to that of the PM6:BTA-C6 blend films (with a π-π stacking distance of 3.53 Å and CCL of 19.67 Å). The closer and more ordered π-π stacking within the PM6:BTA-E3 blend film could be attributed to the intermolecular interaction facilitated by the H-bonding of the ethyl ester side chain. Unexpectedly, the disparate diffraction patterns have arisen within the PM6:BTA-E6 and PM6:BTA-E9-based blend films. Regrettably, the superimposition of peaks at similar positions poses challenges to the quantitative analysis. It is evident that a unique triplet peak at 1.49-2.11 Å⁻¹ along the OOP direction was observed in PM6:BTA-E6 and PM6:BTA-E9 blend films and the π-π stacking distances were evaluated to be approximately 3.37 Å, 3.60 Å and 3.90 Å, indicating the distinctive π-π molecular stacking behaviors within the blend films. In addition, from BTA-E3 to BTA-E6 and then to BTA-E9, the (100) peak of lamellar stacking within PM6 (0.30 Å⁻¹ along IP direction) and acceptors (0.43 Å⁻¹ along IP direction) exhibited enhanced sharpness and definition, implying the enhanced respective crystallization behavior of donor and acceptor components. Moreover, the PM6:BTA-E6 and PM6:BTA-E9 blend films exhibited the gradually clear high-order crystallization (200) peak at 0.59 Å⁻¹ along the IP direction[22,35,57], which corresponds to the lamellar stacking of PM6, while the PM6:BTA-E3 and PM6:BTA-C6 blend films showed a relatively indistinct (200) peak. The aforementioned observation suggests that as the ethyl ester side

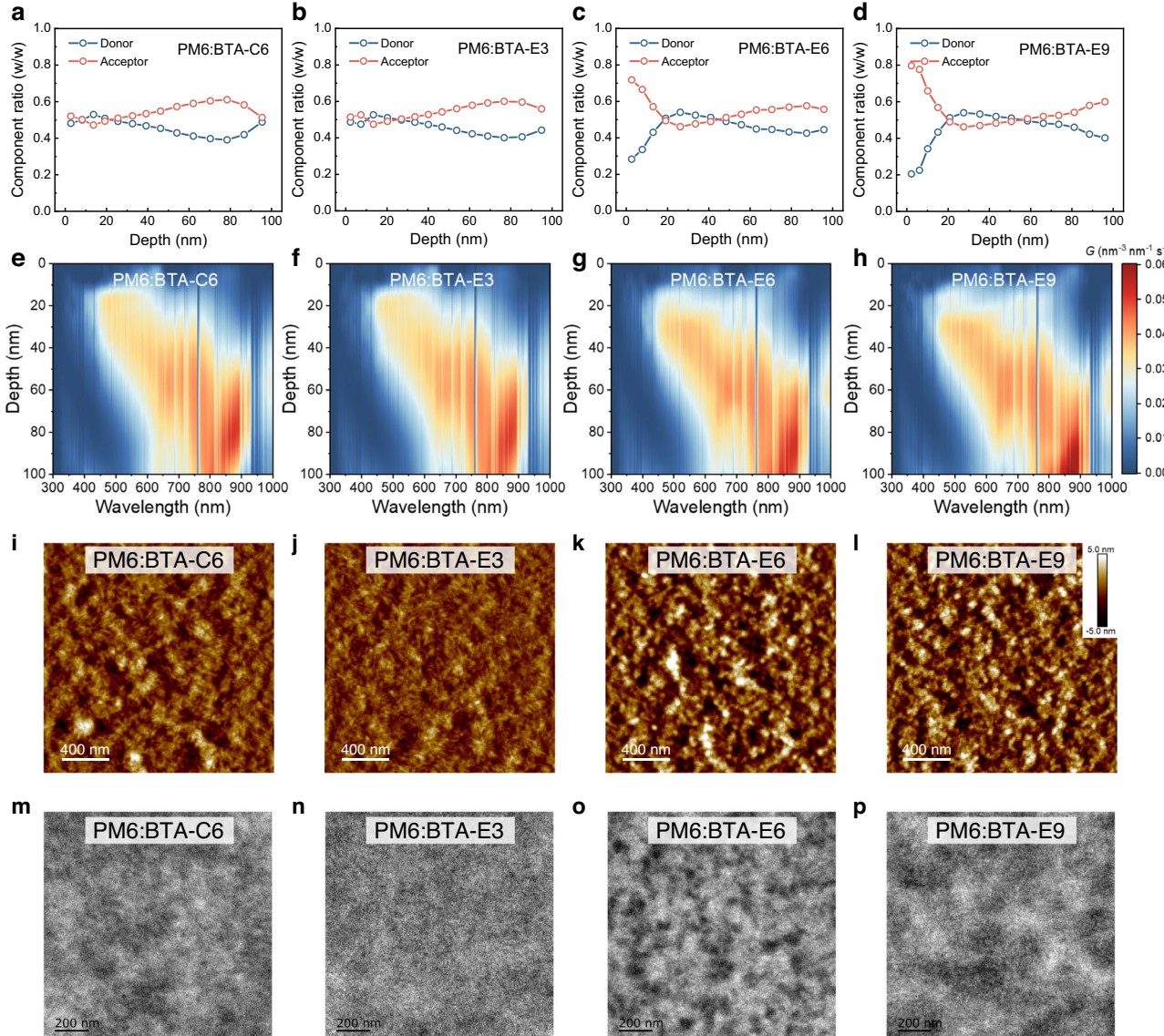

**Fig. 5 | Vertical phase distribution and morphology of the blend films.**
**a–d** Components distribution profiles of the blend films at different film-depths.
**e–h** Numerical simulations for the exciton generation contours. **i–l** AFM height images of the blend films. **m–p** TEM images of the blend films. Source data are provided as a Source Data file.

chain elongates, the miscibility between the donor and acceptor decreases, which may lead to the increased orientation of lamellar stacking of donor and acceptor within the blend films. The proper length ethyl ester side chain of BTA-E3 could promote the close and ordered molecular stacking, thereby enhancing the charge transport and photovoltaic performance of the OSCs.

Film-depth-dependent light absorption spectroscopy (FLAS) was conducted to investigate the distribution of donor and acceptor at the different depths within the films (Supplementary Fig. 48)[58,59]. As depicted in Fig. 5a–d, an uneven vertical phase distribution was observed in the blend films of PM6:BTA-E6 and PM6:BTA-E9, accompanied by the steep donor and acceptor component ratio curves and a significantly higher content of acceptor component compared to the donor at the top interface. In contrast, the ratios of donor and acceptor components within different depths in the PM6:BTA-E3 and PM6:BTA-C6 blend films were closer to their initial blend ratio. Besides, the PM6:BTA-E3 blend films exhibited a much flatter component ratio curve, indicating a more uniform vertical phase distribution that could facilitate the hole and electron transport. Consequently, the exciton

generation contours and the exciton generation rate (*G*) curves were obtained through the numerical simulation of the transfer matrix method from the FLAS profiles[60,61]. As illustrated in Fig. 5e–k and Supplementary Fig. 49, the exciton in the PM6:BTA-E6 and PM6:BTA-E9 blend films mainly generated near the bottom interface, with the maximal exciton generation rate located at approximately 60 nm. This means that the majority of free electrons need to traverse half of the active-layer in order to reach the cathode and be collected, thereby increasing the potential risk of charge recombination due to their long transport distance. In comparison, the maximal population of exciton generation for the PM6:BTA-E3 and PM6:BTA-C6 blend films occurred in the middle regions of the active layer, which could facilitate more efficient hole and electron transport and achieve high $J_{sc}$ and FF. In addition, to illustrate the PCE enhancement of 2PACz-based devices, the vertical phase distribution of the PM6:BTA-E3 blend films was investigated when different hole transport layers were employed. Notably, a favored vertical phase distribution was observed in the 2PACz-based devices, wherein the donors were enriched at the lowermost regions of the film (Supplementary Fig. 50). Besides, the

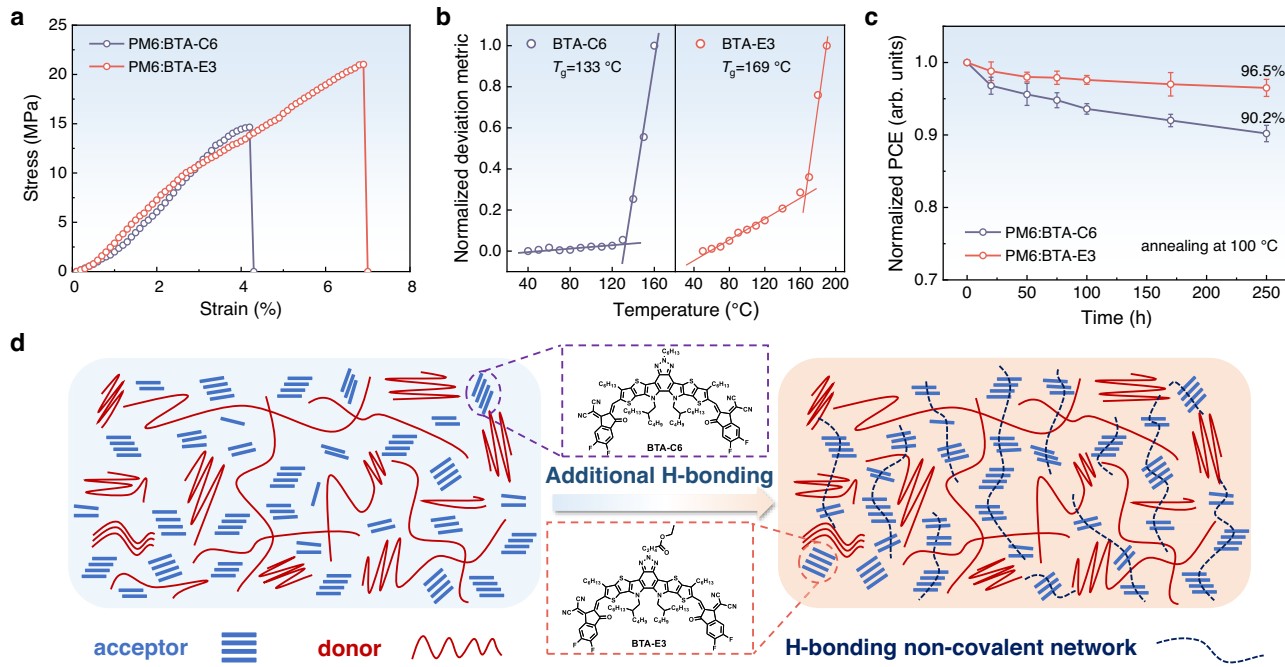

**Fig. 6 | Mechanical properties and stability. a** Stress-strain curves of the PM6:BTA-C6 and PM6:BTA-E3 blend films by film-on-water measurement. **b** $T_g$ of BTA-C6 and BTA-E3. **c**, Normalized PCE of the OSCs based on PM6:BTA-C6 and PM6:BTA-E3 during annealing at 100 °C for 250 h. Error bars represent the standard error of the mean ($n = 10$). **d** Diagram of H-bonding non-covalent cross-link network within blend films. Source data are provided as a Source Data file.

calculated $G$ values of PM6:BTA-E3 blend film upon 2PACz substrate were higher than its PEDOT:PSS counterpart. The more favorable vertical phase separation and more generated excitons could mainly account for the enhanced $J_{sc}$ and FF of the 2PACz-based OSCs.

The miscibility of PM6 and the BTA-series acceptors was further investigated by measuring the contact angle of the neat films (Supplementary Fig. 51). The surface energy of BTA-E3, BTA-C6, BTA-E6, BTA-E9 and PM6 were determined as 36.75, 37.53, 38.63, 40.53 and 32.96 mN m$^{-1}$, respectively. The Flory-Huggins interaction parameters ($\chi$) could be obtained from the equation: $\chi = K(\sqrt{\gamma_D} - \sqrt{\gamma_A})^2$, where $K$ is a constant, $\gamma_D$ and $\gamma_A$ are the surface energy of donor and acceptor[62]. The $\chi$ values of the blend films of PM6:BTA-C6, PM6:BTA-E3, PM6:BTA-E6, and PM6:BTA-E9 were calculated to be $0.15\,K$, $0.10\,K$, $0.22\,K$, and $0.39\,K$, respectively. Compared to BTA-C6, BTA-E3 demonstrated the better miscibility with PM6, as evidenced by the smaller $\chi$ value, suggesting that the PM6:BTA-E3 blend film could achieve a higher density of D/A interface, thus facilitating the efficient photogenerated exciton dissociation. Additionally, the $\chi$ values of BTA-E6 and BTA-E9 increased from $0.22\,K$ to $0.39\,K$, indicating their poorer miscibility with PM6 as also observed in the GIWAXS measurement.

Furthermore, atom force microscopy (AFM) and transmission electron microscopy (TEM) were used to investigate the surface morphology and phase separation within the PM6:BTA-series-based blend films. As illustrated in Fig. 5i–l, each blend film exhibited different surface morphology, and the root mean square (RMS) roughness of the BTA-C6, BTA-E3, BTA-E6, and BTA-E9 based blend films were determined to be 1.22 nm, 0.961 nm, 1.74 nm, and 1.69 nm. Particularly, there is a well-intermixed morphology characterized by a smooth surface and an apparent fiber-like bicontinuous network observed in the PM6:BTA-E3 and PM6:BTA-C6 blend films, which could facilitate the exciton dissociation and charge transport. However, the PM6:BTA-E6 and PM6:BTA-E9 blend films exhibited a rough surface with significant fluctuations, which can be attributed to the excessive aggregation of acceptor molecules as well as the poorer miscibility between the donor and acceptor components. The phase distribution was further observed through the TEM images of blend films (Fig. 5m–p). All

the blend films exhibited a uniform phase separation and nanoscale bicontinuous network, with the exception of the PM6:BTA-E9 blend film. Specifically, the high densities and uniform distribution of dark crystallites in the PM6:BTA-E3 blend films indicated the small domain scale and homogenous phase distribution. In contrast, the PM6:BTA-C6 and PM6:BTA-E6 blend films exhibited relatively larger phase sizes, while the PM6:BTA-E9 blend films demonstrated significant block-like phase separation. It is noteworthy that the dark spot signal observed in the PM6:BTA-E6 and PM6:BTA-E9 blend films exhibited significantly enhanced intensity, indicating the intense aggregation of acceptor molecules, which could serve as the primary factor contributing to the degradation of device performance. The optimal domain size and homogenous phase separation along with the fiber-like bicontinuous network structure within the PM6:BTA-E3 blend film confer significant advantages for efficient exciton dissociation, enhanced carrier transport, and effective suppression of recombination losses, thereby contributing to the improved device performance.

## Mechanical properties and device stability
As previously mentioned, the compelling evidence unequivocally validated that the photovoltaic performance of the OSCs based on BTA-E3 was significantly enhanced through the introduction of dynamic H-bonding via ethyl ester side chain, compared to the BTA-C6 counterpart. Furthermore, in order to delve into the effect of dynamic H-bonding on the improvement of mechanical properties of the active-layer films, the film-on-water (FOW) measurement was carried out for the PM6:BTA-C6 and PM6:BTA-E3 blend thin films[24,63]. The mechanical properties of fracture strain, elastic modulus, and toughness could be derived from the stress-strain curves depicted in Fig. 6a. Therein, the COS is obtained from the corresponding strain at specimen failure, and the elastic modulus is derived from the slope of the elastic region within the stress-strain curve. The COS values for PM6:BTA-C6 and PM6:BTA-E3 blend films were determined to be 4.3% and 7.0%, with their respective elastic moduli measuring at 453 MPa and 404 MPa. Incorporating the ethyl ester side chain, the PM6:BTA-E3-based blend films exhibited an improved COS and a decreased elastic modulus

compared to the PM6:BTA-C6 counterpart, indicating the enhanced tensile strength and mechanical robustness of the films. Moreover, the toughness, derived from the enclosed area under the stress-strain curves, of the PM6:BTA-C6 and PM6:BTA-E3 blend films were determined to be 0.29 MJ m⁻³ and 0.79 MJ m⁻³ respectively. It is worth noting that the toughness of the PM6:BTA-E3 blend film exhibited an approximately twofold increase compared to that of the PM6:BTA-C6 film. In addition, as the ethyl ester side chain length elongating, the blend films of PM6:BTA-E6 and PM6:BTA-E9 exhibited gradually increased toughness of 0.83 MJ m⁻³ and 0.94 MJ m⁻³ respectively (Supplementary Fig. 52). The enhancement in toughness unequivocally signifies an enhanced capacity to withstand crack and deformation. In a nutshell, the dynamic H-bonding undeniably harbors immense potential in augmenting the stretchability and toughness of the films.

In order to investigate the impact of dynamic H-bonding on the photovoltaic performance and mechanical robustness of OSCs, we fabricated the flexible devices with the structure of PET/ITO/2PACz/ Active layer/PDINN/Ag. Considering that BTA-E3 has the highest device efficiency and a similar molecular structure to BTA-C6, we selected BTA-E3 and BTA-C6 as the acceptor materials of active-layer to more intuitively reflect the effect of H-bonding modification on the mechanical durability of flexible OSCs. The $J$-$V$ curves of the flexible OSCs are shown in Supplementary Fig. 53a and the corresponding photovoltaic performance parameters are listed in Supplementary Table 6. Notably, the PM6:BTA-E3-based devices exhibited a PCE of 18.33% with the $V_{oc}$ of 0.829 V, $J_{sc}$ of 28.18 mA cm⁻², and FF of 78.38%. While the PM6:BTA-C6-based devices only achieved a PCE of 17.18%. Moreover, the mechanical durability of these flexible OSCs was tested at bending radium of 5 mm. As depicted in Supplementary Fig. 53b, the PM6:BTA-E3-based flexible devices maintained 88.91% of the initial PCE after 2000 consecutive bends, while the PCE of PM6:BTA-C6-based devices rapidly decayed to 77.85%. The better device performance and mechanical endurance of the PM6:BTA-E3-based flexible devices imply that the dynamic H-bonding modification could effectively and simultaneously improve the performance and durability of the flexible OSCs.

In addition to enhance the mechanical properties, the formed non-covalent inter-chain cross-linked network through intermolecular dynamic H-bonding also holds promise in improving the stability of the active-layer blend film morphology. The intricate microstructure within the blend film is regarded as a thermodynamically metastable system. When exposed to illumination, the elevation in temperature could prompt the diffusion and migration of both donor and acceptor within the blend film, thus destroying the morphology of the active-layer and affecting the performance of the devices. Therefore, inhibition of photovoltaic materials diffusion plays a crucial role in the long-term stability of OSCs. To comprehend the diffusion behavior of BTA-E3 and BTA-C6, the glass transition temperature ($T_g$) was determined by analyzing the deviation in absorbance between as-cast films and annealed films at different temperatures[64]. Notably, BTA-E3 exhibited a much higher $T_g$ of 169 °C compared to its counterpart BTA-C6 with a $T_g$ of 133 °C (Fig. 6b), suggesting that the existence of the ethyl ester side chain could enhance the intermolecular H-bonding non-covalent interactions. Furthermore, the diffusion coefficient at 85 °C ($D_{85}$) of the PM6-based blend films could be calculated from their $T_g$[65], and the corresponding $D_{85}$ was calculated to be $4.18 \times 10^{-20}$ cm² s⁻¹ for PM6:BTA-C6 and $1.89 \times 10^{-22}$ cm² s⁻¹ for PM6:BTA-E3. The higher $T_g$ and lower diffusion coefficient could be advantageous in constraining the chain movement and maintaining the blend film morphology, thereby enhancing the stability of OSCs based on PM6:BTA-E3. Moreover, the PM6:BTA-C6 and PM6:BTA-E3 blend films were continuously annealed at 100 °C in the nitrogen glove box for various durations to compare their thermal stability. Impressively, the PM6:BTA-E3-based OSCs demonstrated enhanced thermal tolerance in comparison to their

PM6:BTA-C6 counterpart. As illustrated in Fig. 6c, the PM6:BTA-E3-based OSCs maintained 96.5% of the initial PCE after annealing at 100 °C for 250 hours, surpassing significantly the parallel case of the PM6:BTA-C6 device (90.2%). The PCE decay of the PM6:BTA-E6 and PM6:BTA-E9-based OSCs under thermal stress was also enhanced with the respective PCE retention of 95.4% and 93.7% after annealing (Supplementary Fig. 54a). In addition, the PM6:BTA-E3 based OSCs also exhibited remarkable long-term stability storing in the nitrogen glove box (Supplementary Fig. 54b), and even under high humidity condition (Supplementary Fig. 54c–d). The results imply that incorporating the dynamic H-bonding intermolecular interaction into the molecular design may be a very effective strategy to promote the formation of non-covalent cross-linked networks between photovoltaic materials (Fig. 6d) and significantly improve both the resistance against the thermal stress dissipation and the morphological stability.

## Discussion

In summary, with the aim of simultaneously enhancing the photovoltaic performance and mechanical robustness of OSCs, the dynamic H-bonding ethyl ester side chains were incorporated into the molecule design of SMAs. We synthesized a series of BTA-based acceptors with different side chains on the BTA core unit, namely BTA-C6 with $n$-hexyl side chain, and BTA-E3, BTA-E6 and BTA-E9 with different carbon length ethyl ester side chains, and systematically investigated the influence of dynamic H-bonding and the flexible side chain length on the photoelectric properties, crystallization behaviors and photovoltaic performances. The different side chains on the BTA central core have an insignificant impact on energy levels. While as ethyl ester side chains elongate from BTA-E3 to BTA-E6 and then to BTA-E9, their miscibility with PM6 is getting poorer, which will cause excessive aggregation and improper phase vertical separation. Consequently, the PM6:BTA-E6 and PM6:BTA-E9-based OSCs exhibited relatively larger energy losses and poorer device performances. Due to the synergetic effect of H-bonding modification and side-chain engineering, BTA-E3 demonstrated higher charge carrier generation, less charge recombination, and more rapid charge transport. Single-crystal X-ray diffraction analysis indicates BTA-E3 demonstrates strong H-bonding interactions and could form ordered molecular stacking and dense 3D charge transport channels. Combined with the optimization of hole transport layer, the PM6:BTA-E3 based binary OSCs with 2PACz as the hole transport layer and processed by $o$-xylene non-halogenated solvent achieved an extraordinary PCE of 19.92% (certified as 19.57%) with an FF of 80.63%, setting a record PCE in green solvents processed binary OSCs. Significantly, the PM6:BTA-E3 blend films also realized an improved COS value of 7.0% and enhanced toughness of 0.79 MJ m⁻³, dramatically surpassing the PM6:BTA-C6 counterpart without the ethyl ester side chain. The PM6:BTA-E3-based flexible OSCs achieved a remarkable PCE of 18.33% and exhibit enhanced mechanical durability than that of PM6:BTA-C6-based devices. Furthermore, the PM6:BTA-E3-based system also exhibited better stability, maintaining 96.5% of its initial efficiency after heating at 100 °C for 250 hours. Our investigations reveal that the manipulation of dynamic H-bonding interaction bears immense significance in improving the photovoltaic performance, mechanical robustness, and device stability, offering innovative perspectives for the design and advancement of flexible photovoltaic materials.

## Methods

### Materials and synthesis
Polymer PM6 was purchased from Solarmer Materials Inc. The 4,7-dibromo-2H-benzo[d][1,2,3]triazole was purchased from Nanjing Zhi-yan Technology Ltd and the ethyl 3-bromopropanoate was purchased from J&K Scientific. Other chemical reagents and solvents were purchased from commercial sources and used as received. Compound BTA-C6 was synthesized according to the procedures reported in the

previous literature[66]. The detailed synthetic routes and procedures of BTA-E3, BTA-E6, and BTA-E9 are shown in Supplementary Information.

## Molecular structure characterization

$^1$H NMR and $^{13}$C NMR spectra were recorded using Bruker Fourier 300 and Bruker AV 400 spectrometer in *d*-chloroform solution. Chemical shifts were reported as δ values (ppm) with tetramethyl silane (TMS) as reference. High-resolution matrix-assisted laser desorption ionization-time of flight mass spectrometry (MALDI-TOF MS) was performed on the Shimadzu spectrometer.

## Thermogravimetric analysis

Thermogravimetric analysis (TGA) was performed by PerkinElmer TGA8000 under a nitrogen atmosphere with 20 °C min$^{-1}$ heating rate from 50 °C to 500 °C.

## UV-vis absorption

UV-vis absorption spectra were recorded on the Hitachi UH5700 UV-vis spectrophotometer. For solution absorption, the materials were dissolved in *o*-xylene. For the film absorption, the corresponding *o*-xylene solutions were spin-coated on quartz plates.

## Cyclic voltammetry

The cyclic voltammetry (CV) measurement was performed on the CHI660C electrochemical workstation, using a glassy carbon electrode as the working electrode, platinum wire as the counter electrode and Ag/AgCl as the reference electrode, at a potential scanning rate of 50 mV s$^{-1}$ in 0.1 M tetrabutylammonium hexafluorophosphate (Bu$_4$NPF$_6$) acetonitrile solution. The ferrocene/ferrocene (Fc/Fc$^+$) pair was used as internal reference and the energy levels were calculated using the following formula: $E_{HOMO}/E_{LUMO} = -e(\varphi_{ox}/\varphi_{red} + 4.8 - \varphi_{Fc/Fc^+})$ (eV).

## Single-crystal measurement

Single-crystal of BTA-E3 was cultivated using a solvent diffusion method with isopropanol as a poor solvent and chloroform as a good solvent. According to standard procedure, X-ray single-crystal data is collected on XtaLAB Synergy-R at low temperature protected by liquid nitrogen, analyzed using Mercury (version 4.0), and deposited at the Cambridge Crystallographic Data Center (CCDC).

## Fabrication and characterization of OSCs

All the OSCs were fabricated with the device structure of ITO/PEDOT:PSS(2PACz)/active layer/PDINN/Ag. The pre-patterned ITO glasses substrate (sheet resistance = 15 Ω sq$^{-1}$) were sonicated sequentially in detergent, deionized water, acetone, and isopropanol. The glasses were dried in a vacuum oven and treated by ultraviolet ozone (Jelight Company, USA) for 30 minutes. For the device using PEDOT:PSS as hole transport layer, PEDOT:PSS aqueous solution (Baytron P 4083, from HCStarck) was filtered through a 0.45 mm filter, and spin-coated on the pre-cleaned ITO glasses at 6000 rpm for 30 seconds, and then the ITO glasses were dried in the air at 150°C for 30 minutes. The PEDOT:PSS coated ITO substrates were transferred to N$_2$-filled glove box for further processing. For the device using 2PACz as the hole transport layer, the isopropanol solution of 2PACz with a concentration of 0.5 mg mL$^{-1}$ was spin-coated on the pre-cleaned ITO glasses at 3000 rpm for 30 seconds, followed by annealing at 100°C for 10 minutes in N$_2$-filled glove box. The PM6:SMAs (D:A = 1:1.2) was dissolved in *o*-xylene (19 mg mL$^{-1}$ in total) and stirred at 60°C for at least 2 hours. An additive, 1,8-diiodooctane (DIO) (volume content: 0.3%) was added into the solution half an hour before spin coating. The blend solution was spin-coated on the PEDOT:PSS (2PACz) coated ITO substrate at 3000 rpm for 30 seconds, followed by annealing at 100°C for 10 minutes. After cooling down to room temperature, PDINN methanol solution with a concentration of 1.0 mg mL$^{-1}$ was spin-coated on the active layers at 3000 rpm for 30 seconds. Then all the samples were

transferred to the evaporation chamber. Under a vacuum of $2 \times 10^{-6}$ Pa, Ag electrode (100 nm) was deposited through a shadow mask, and the device area is 6.0 mm$^2$.

The current density-voltage (J-V) characteristic of OSCs was measured in the nitrogen-filled glove box equipped with Keysight B2901A, using Newport 94023 A with 450 W xenon lamp and AM 1.5 G filter as the light source. The light intensity was calibrated to 100 mW cm$^{-2}$ by Enlitech SRC2020. The external quantum efficiency (EQE) of OSCs was measured by Enlitech QE-RT3011. The light intensity at each wavelength was calibrated by standard single-crystal silicon photovoltaic cells.

## Charge dynamics characterization

The data of transient photocurrent (TPC), transient photovoltage (TPV), and photoinduced charge carrier extraction by linearly increasing voltage (Photo-CELIV) were obtained by the all-in-one characterization platform, Paios (Fluxim AG, Switzerland). In TPC testing, the light intensities were 10%, 17.8%, 31.6%, 56.2%, and 100% sunlight, respectively. The settling time was 100 μs, pulse length was 100 μs and the follow-up time was 200 μs. In the TPV testing, the light intensities were 0.10%, 0.23%, 0.53%, 1.23%, 2.83%, 6.52%, 15.0%, 34.6%, and 80.0% sunlight, respectively. The settling time was 30 ms, pulse length was 5 ms and the follow-up time was 30 ms. In the Photo-CLIVE testing, the delay time was set to 0 s, the light intensity was 100% sunlight, the light-pulse length was 100 μs, and the sweep ramp rate was raised from 20 V ms$^{-1}$ to 100 V ms$^{-1}$.

## Mobility measurements

The electron and hole mobility were measured by using the method of space charge limited current (SCLC), where ITO/PEDOT:PSS/active layer/MoO$_3$/Ag device structure was used to test hole mobility and ITO/ZnO/active layer/PDINN/Ag device structure was used to test electron mobility. The hole and electron mobilities were calculated according to the equation: $J = 9\mu\varepsilon_r\varepsilon_0 V^2/8d^3$, where $J$ is the current density, $\mu$ is the hole or electron mobility, $V$ is the internal voltage in the device, $\varepsilon_r$ is the relative dielectric constant of active-layer materials, $\varepsilon_0$ is the permittivity of empty space and $d$ is the thickness of the active-layer.

## Energy loss measurements

Fourier-transform photocurrent spectroscopy external quantum efficiency (FTPS-EQE) was measured using an integrated system (PECT600, Enlitech), where the photocurrent was amplified and modulated by a lock-in instrument. External electroluminescence quantum efficiency (EQE$_{EL}$) measurements were performed by applying external voltage/current sources through the devices (ELCT−3010, Enlitech). All the devices were fabricated for EQE$_{EL}$ measurements according to the optimal device fabrication conditions.

## Transient absorption spectroscopy

Femtosecond transient absorption spectrometer was composed of a regenerative-amplified Ti: sapphire laser system (Coherent) and Helios pump-probe system (Ultrafast Systems). The regenerative-amplified Ti: sapphire laser system (Legend Elite-1K-HE, center wavelength of 800 nm, pulse duration of 25 fs, pulse energy of 4 mJ, repetition rate of 1 kHz) was seeded with a mode-locked Ti: sapphire laser system (Vitara) and pumped with a Nd: YLF laser (Evolution 30). The output 800 nm fundamental of the amplifier was split into two beam pulses. The main part of the fundamental beam went through the optical parametric amplifiers (TOPAS-C), whose output light was set as the pump light with a wavelength of 830 nm and chopped by a mechanical chopper operating at a frequency of 500 Hz. A small part of the fundamental beam was introduced into the TA spectrometer in order to generate the probe light. After passing through a motorized optical delay line, the fundamental beam was focused on a sapphire crystal or

YAG crystal, which was used to generate the white-light continuum (WLC) probe pulses with wavelengths of 430 to 820 nm or 800 to 1600 nm, respectively. The optical path difference between the pump light and the probe light, which is controlled by the motorized optical delay-line, was used to monitor the transient states at different pump-probe delays. A reference beam was split from the WLC in order to correct the pulse-to-pulse fluctuation of the WLC. The pump was spatially and temporally overlapped with the probe beam on the sample. The excitation energy of the pump pulse was set to 2 μJ/cm2 to avoid singlet-singlet annihilation. The film samples for TA measurements were prepared by spin-coating the corresponding materials on thin quartz plates. The film samples were thermally annealed the same way as the actual device.

### Grazing incidence wide-angle X-ray scattering (GIWAXS) measurements

GIWAXS measurements were performed at beamline 7.3.3 at the Advanced Light Source[67]. Samples were prepared on Si substrates using identical blend solutions as those used in devices. The 10 keV X-ray beam was incident at a grazing angle of 0.14°, selected to maximize the scattering intensity from the samples. The scattered X-rays were detected using a Dectris Pilatus 2 M photon counting detector. 1D profile was obtained with the intensity distribution analyzed along in-plane and out-of-plane directions. Crystal coherence lengths (CCL) are estimated based on the Scherrer equation (CCL = 2πK/FWHM), where K is the shape factor (here we use 0.9), and FWHM is the full width at half maximum of diffraction peaks.

### Film-depth-dependent light absorption spectroscopy

The Film-depth-dependent light absorption (FLAS) spectra were acquired upon a film-depth-dependent light absorption spectrometer (PU100, Puguangweishi Co. Ltd.). In-situ oxygen plasma etching at low pressure was used to extract the depth-resolved absorption spectrum for the active layer. From the evolution of the spectra and Beer-Lambert's Law, film-depth-dependent absorption spectra were extracted. The exciton generation contour is numerically simulated upon inputting sublayer absorption spectra into a modified optical transfer-matrix approach.

### AFM and TEM

The atomic force microscopy (AFM) images were collected on Bruker ICON2-SYS in tapping mode with a 2 μm scanner, and the transmission electron microscope (TEM) measurements were performed on JEOL JEM-1400. All the blend films were spin-coated on ITO/PEDOT: PSS substrate with 10 minutes of annealing at 100 °C.

### Mechanical measurements

Stress-strain curves were acquired using a custom-designed film-on-water (FOW) instrument. In the FOW test, the blend films were coated on precleaned glasses (size: 2 × 2 cm) and then transferred to the water surface. The blend films were moved above the PDMS fixture and then glued to the PDMS by lowering the liquid level. The thickness of the blend films is about 100 nm.

### Reporting summary

Further information on research design is available in the Nature Portfolio Reporting Summary linked to this article.

## Data availability

The data supporting the findings of this study are available within the published article and Supplementary Information and Source Data Files. Additional data are available from the corresponding author on request. The X-ray crystallographic coordinates for structures reported in this study have been deposited at the Cambridge Crystallographic Data Centre (CCDC), under deposition number 2333735. Source data are provided in this paper.

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

## Acknowledgements

This work was supported by the Beijing Nova Program (20240484597) (X.L.) , the National Natural Science Foundation of China (Nos. 52203248 (X.L.), 52103243 (J.Z.) and 52173188 (L.M.)), the Key Research Program of the Chinese Academy of Sciences (No. XDPB13) (J.H.) and the Basic and Applied Basic Research Major Program of Guangdong Province (No. 2019B030302007) (Y.L.), and the Strategic Priority Research Program of the Chinese Academy of Sciences (No. XDB0520102) (L.M.).

## Author contributions

X.L. conceived and directed this project. H.H. designed and synthesized the BTA-series acceptors, and fabricated and characterized the OSCs based on these acceptors. Z.C., H.Z., and Y.C.L. participated in the optimization of device fabrication. Y.G. grew the single-crystal of BTA-E3, and T.L. solved and analyzed the single-crystal structures. X.W. contributed to the AFM and TEM measurements. J.Z. conducted the TA measurements and data analysis. S.W., Z.B., and W.M. carried out the GIWAXS measurements and assisted with data analysis. B.S. and G.L. conducted the FLAS characterization and analysis. K.Z. and L.Y. measured and analyzed the FOW test. B.Z. and Y.W.L fabricated and characterized the flexible devices. H.H. and X.L. wrote the manuscript and Y.F.L., X.L., and L.M. contributed to revisions of the manuscript. All the authors participated in the data analysis and commented on the manuscript.

## Competing interests

The authors declare no competing interests.
