## [Peer Review File · Nature Communications]

REVIEWER COMMENTS

Reviewer #1 (Remarks to the Author):

This manuscript describes a comparative study of small molecule acceptors (SMAs) with ester-containing side chains. The authors show the presence of an H-F hydrogen bonding interaction between the alpha-hydrogen of the ester side chain and the fluorinated end groups of the SMAs in the case of one of the SMAs (BTA-E3). When the alkyl spacer between the SMA and the ester is an optimal length (2 carbons) in the case of BTA-E3, the OPV performance and lifetime as well as mechanical properties in blends with the well-known donor polymer PM6 are enhanced relative to the SMA with an alkyl chain of 6-carbons (BTA-C6). The authors provide extensive characterization on the comparison of BTA-E3 and BTA-C6. The performance and stability of the solar cells as well as the mechanical properties are a strength of the work.

There are a few weaknesses of the work:

1. The authors claim this is the highest reported PCE for a binary organic solar cell processed with a non-halogenated solvent. However, the authors do use 1,8-diiodooctane in the processing of the films. As such, the films are not processed in a halogen-free processing method.
2. In reference to supplementary Figure 36, the authors refer to the two different “configurations” of the ester side chain. This should be corrected to “conformations.”
3. While it is clear that PM6:BTA-C6 and PM6:BTA-E3 were selected for mechanical analysis based on higher solar cell PCE, the exclusion of blends containing BTA-E6 and BTA-E9 is a missed opportunity to demonstrate the impact of H-bonding on mechanical properties as a general consequence of H-bonding. Both BTA-E6 and BTA-E9 have the same motif capable of H-Bonding as BTA-E3. This data would go a long way to elucidate the general impact of this H-Bonding interaction. As such, this data should be added.
4. Similarly, if it is H-Bonding that is the cause of increased thermal stability in organic solar cells, why omit the blends with BTA-E6 and BTA-E9? Regardless of the starting PCE, this offers an opportunity to show the impact of this H-Bonding motif on stability. This data should be added.

Overall, this manuscript reports interesting findings. More complete analysis of the BTA-E6 and BTA-E9 SMAs for thermal stability in organic solar cells and mechanical properties will make this a strong paper. Without these comparisons the impact of the work is limited as it can be argued the results are due to one specifically optimal SMA and not a general concept.

Reviewer #2 (Remarks to the Author):

This work reported synthesis of a series of BTA-based acceptors with different side chains using BTA core unit, BTA-C6, BTA-E3, BTA-E6, and BTA-E9. The authors studied effects of H-bonding and the flexible side chain length on the photovoltaic properties and the PM6:BTA-E3 based solar cell achieved PCE of 19.92% when being processed by o-xylene non-halogenated solvent. There are several critical issues to be addressed.

- It is important to demonstrate why PM6:BTA-E3 shows optimal morphology and high performance when being prepared with o-xylene non-halogenated solvent. What are the critical characteristics of BTA-E3 (or PM6:BTA-E3) which make the BHJ appropriate for the solution process using non-halogenated solvents compared with other acceptors reported so far. What are the effects of the hydrogen-bonding to make nonhalogenated solvent-based process favorable? The authors need to address these issues by mechanism studies of morphology evolution during coating.
- How about the trend of photovoltaic properties among BTA-C6, BTA-E3, BTA-E6, and BTA-E when the device is prepared using halogenated solvents? Please, explain the results and not just describe the result. It is necessary to explain the relationship between chemical structures of the acceptors and properties of the solvents.
- It is already known that non-covalent bonding such as hydrogen bonding is favorable for mechanical stability of the BHJ film. How about the stability of the BHJ film under high humidity? It is possible that the hydrogen bonding of the acceptor makes the BHJ more hydrophilic and allows the water molecules to easily diffuse inside.

Reviewer #3 (Remarks to the Author):

This manuscript is about introducing hydrogen bonding to enhance stretchability and achieve remarkable efficiencies of OPVs. However, the concept of this study is not sufficiently novel to be published in "Nature Communications". This approach is well known in the organic semiconductor community (e.g. Chem. Mater. 2020, 32, 13, 5700–5714, Adv. Mater. 2022, 34, 2207544, Chem. Mater. 2023, 35, 24, 10476–10486). Additionally, the crack onset strain of active layer system with polymer donor and small molecule acceptor have been improved to over 13% (e.g. J. Am. Chem. Soc. 2023, 145, 11914–11920), almost doubling the COS value reported in this paper. Furthermore, while the authors claim mechanical robustness for organic solar cells, they lack measurements for the mechanical durability of flexible/stretchable OPVs. Therefore, I recommend submitting this manuscript to more specialized journals rather than "Nature Communications".

I also include comments to improve this manuscript.

- (1) I am curious about the hydrogen bonding C-H...F. Based on the chemical structure, the other materials demonstrate the same H bonding units, could we observe the H bonding in the other materials? Please discuss this in detail.
- (2) How about the H-bonding strength in BTA-E3?
- (3) How do we understand the effect of H-bonding strength on devices' thermal stability? H-bonding is thermodynamically reversible, is the thermal stability also reversible?
- (4) The authors mentioned that the 2PACz HTL is better than PEDOT:PSS, leading to the new record PCE. The detailed discussions of the PCE enhancement are lacking.
- (5) The authors claimed that this OPV shows a new recorded PCE in binary devices processed with non-halogenated solvents. The authors should provide references to compare the efficiency.
- (6) Please specify the statistic number when demonstrated the error bars in the figures.
- (7) Please specify the full name of PM7.

Response to referees

Response to Referee #1:

This manuscript describes a comparative study of small molecule acceptors (SMAs) with ester-containing side chains. The authors show the presence of an H-F hydrogen bonding interaction between the alpha-hydrogen of the ester side chain and the fluorinated end groups of the SMAs in the case of one of the SMAs (BTA-E3). When the alkyl spacer between the SMA and the ester is an optimal length (2 carbons) in the case of BTA-E3, the OPV performance and lifetime as well as mechanical properties in blends with the well-known donor polymer PM6 are enhanced relative to the SMA with an alkyl chain of 6-carbons (BTA-C6). The authors provide extensive characterization on the comparison of BTA-E3 and BTA-C6. The performance and stability of the solar cells as well as the mechanical properties are a strength of the work. There are a few weaknesses of the work:

1. *The authors claim this is the highest reported PCE for a binary organic solar cell processed with a non-halogenated solvent. However, the authors due use 1,8-diiodooctane in the processing of the films. As such, the films are not processed in a halogen-free processing method.*

Response: Thanks for the reviewer's comments. 1,8-diiodooctane (DIO) was utilized as the additive to optimize the film morphology, with a volume of addition of only 0.3%, which accounts for a very small amount in the entire solvent system. So the impact of additives on the environment is far less than that of solvents. Therefore, utilizing the green solvent for fabricating the OSCs holds greater significance than non-halogenated additive. Anyway, we deleted the description of "*the highest reported PCE for a binary organic solar cell processed with a non-halogenated solvent*" in the sections of "Abstract" and "Discussion", and revised the related sentence to "**the PM6:BTA-E3 based OSCs processed by *o*-xylene solvent achieved a high PCE of 19.92% (certified PCE of 19.57%)**" in the "Abstract".

2. *In reference to supplementary Figure 36, the authors refer to the two different*

“configurations” of the ester side chain. This should be corrected to “conformations.”

Response: Thanks for the reviewer’s suggestion. We have changed the “configurations” into the “conformations” in the manuscript on page 8.

3. While it is clear that PM6:BTA-C6 and PM6:BTA-E3 were selected for mechanical analysis based on higher solar cell PCE, the exclusion of blends containing BTA-E6 and BTA-E9 is a missed opportunity to demonstrate the impact of H-bonding on mechanical properties as a general consequence of H-bonding. Both BTA-E6 and BTA-E9 have the same motif capable of H-Bonding as BTA-E3. This data would go a long way to elucidate the general impact of this H-Bonding interaction. As such, this data should be added.

Response: Thanks for the reviewer’s suggestion. According to the suggestion, the mechanical properties of the PM6:BTA-E6 and PM6:BTA-E9 was investigated through the film-on-water test. As shown in **Supplementary Fig. 49**, The COS values for the PM6:BTA-C6, PM6:BTA-E3, PM6:BTA-E6 and PM6:BTA-E9 blend films were determined to be 4.3%, 7.0%, 9.6% and 8.3%, and their respective elastic moduli were measured at 453 MPa, 404 MPa, 215 MPa and 243 MPa. Incorporating the ethyl ester side chain, the PM6:BTA-E3, PM6:BTA-E6 and PM6:BTA-E9 based blend films exhibited the improved COS values and the decreased elastic moduli compared to their PM6:BTA-C6 counterparts, indicating the enhanced tensile strength and mechanical robustness of the films. Moreover, the toughness of the PM6:BTA-C6, PM6:BTA-E3, PM6:BTA-E6 and PM6:BTA-E9 blend films were determined to be 0.29 MJ m⁻³, 0.79 MJ m⁻³, 0.83 MJ m⁻³ and 0.94 MJ m⁻³ respectively (**Supplementary Fig. 49**). It is worth noting that the toughness of PM6:BTA-E3, PM6:BTA-E6 and PM6:BTA-E9 blend films exhibited an approximate twofold increase compared to that of the PM6:BTA-C6 film. As the ethyl ester side chain length elongating, the blend films of PM6:BTA-E3, PM6:BTA-E6 and PM6:BTA-E9 exhibited gradually increased toughness. The results indicate that the modification of ethyl ester side chain could effectively enhance the stretchability and toughness of the active layer films.

We added the corresponding description in the revised manuscript on Page 22: **“In addition, as the ethyl ester side chain length of the SMAs elongating, the blend films of PM6:BTA-E6 and PM6:BTA-E9 exhibited gradually increased toughness of 0.83 MJ m⁻³**

and 0.94 MJ m^{-3} respectively (Supplementary Fig. 49). The enhancement in toughness unequivocally signifies an enhanced capacity to withstand crack and deformation. In a nutshell, the dynamic H-bonding undeniably harbors immense potential in augmenting stretchability and toughness of the films.”.

Supplementary Fig. 49. a, Stress-strain curves of the PM6:BTA-C6, PM6:BTA-E3, PM6:BTA-E6 and PM6:BTA-E9 blend films by film-on-water measurement. **b,** Toughness of the PM6:BTA-C6, PM6:BTA-E3, PM6:BTA-E6 and PM6:BTA-E9 blend films.

4. Similarly, if it is H-Bonding that is the cause of increased thermal stability in organic solar cells, why omit the blends with BTA-E6 and BTA-E9? Regardless of the starting PCE, this offers an opportunity to show the impact of this H-Bonding motif on stability. This data should be added.

Response: Thanks for the reviewer’s suggestion. We have conducted the thermal stability test of PM6:BTA-E6 and PM6:BTA-E9 blend films with the same condition. As depicted in Fig. 6c and Supplementary Fig. 51, after annealing at 100°C for 250 hours, the PM6:BTA-E3, PM6:BTA-E6 and PM6:BTA-E9 based OSCs maintained 96.5%, 95.4% and 93.7% of their initial PCE, which were higher than that of PM6:BTA-C6 counterparts (90.2%). The results imply that incorporating the dynamic H-bonding into the molecular design could significantly improve the resistance against the thermal stress dissipation of the OSCs.

We added the corresponding description on Page 23: “The PCE decay of the PM6:BTA-E6 and PM6:BTA-E9 based OSCs under thermal stress was also enhanced with the respective PCE retention of 95.4% and 93.7% after annealing (Supplementary Fig.

51a).”.

Supplementary Fig. 51. a, Normalized PCE of the OSCs based on PM6:BTA-E6 and PM6:BTA-E9 during annealing at 100 °C for 250 h. Error bars represent the standard error of the mean (n = 10).

Overall, this manuscript reports interesting findings. More complete analysis of the BTA-E6 and BTA-E9 SMAs for thermal stability in organic solar cells and mechanical properties will make this a strong paper. Without these comparisons the impact of the work is limited as it can be argued the results are due to one specifically optimal SMA and not a general concept.

Response to Referee #2:

This work reported synthesis of a series of BTA-based acceptors with different side chains using BTA core unit, BTA-C6, BTA-E3, BTA-E6, and BTA-E9. The authors studied effects of H-bonding and the flexible side chain length on the photovoltaic properties and the PM6:BTA-E3 based solar cell achieved PCE of 19.92% when being processed by o-xylene non-halogenated solvent. There are several critical issues to be addressed.

1. *It is important to demonstrate why PM6:BTA-E3 shows optimal morphology and high performance when being prepared with o-xylene non-halogenated solvent. What are the critical characteristics of BTA-E3 (or PM6:BTA-E3) which make the BHJ appropriate for the solution process using non-halogenated solvents compared with other acceptors reported so far. What are the effects of the hydrogen-bonding to make nonhalogenated solvent-based process favorable? The authors need to address these issues by mechanism studies of morphology evolution during coating.*

Response: According to the suggestion, the mechanism of morphology evolution during coating of the BTA-E3 based blend film was measured by *in situ* UV-vis absorption spectra, and compared with the widely used and represent acceptors Y6, L8-BO and its counterparts BTA-C6.

We added the corresponding discussion in the Supplementary Note II of Supporting Information: “**Supplementary Note II**”

To elucidate the evolution of the blend film during the coating process, the *in situ* UV-vis absorption spectra of the blend films processed with o-xylene were monitored. The widely used and representative acceptors Y6 and L8-BO were chosen for comparison with BTA-C6 and BTA-E3 to investigate the role of H-bonding in the non-halogenated solvent processed films. As depicted in Supplementary Figs. 53a-d, the film formation process could be divided into three stages. In stage I, the blend solutions were spin-coated on the substrate and the absorption peak location of acceptors remained almost unchanged. In stage II, due to the solvent evaporation, the solution concentration exceeded the solubility limit and caused a rapid red-shift in the absorption peak of acceptors. Finally, upon complete solvent evaporation, the blend film was ultimately formed and both the absorption peak location and intensity reached a constant value (stage III). Among the three stages, the stage II is identified as critical to film evolution as it determines the film morphology.

As shown in Supplementary Figs. 53e-h, the duration of stage II of the PM6:Y6 and

PM6:L8-BO blend films was determined to be 2.9 s and 3.5 s, while the PM6:BTA-C6 and PM6:BTA-E3 blend films exhibited shorter duration of 2.5 s and 2.3 s respectively. The long crystalline time may result in excessive aggregation of Y6 and L8-BO, leading to the unfavorable morphology in the *o*-xylene processed blend films. Moreover, the BTA-E3 exhibited the faster film formation process in comparison with its counterpart BTA-C6. Notably, the absorption peak locations of PM6 in stage I of the PM6:Y6 and PM6:L8-BO solutions were found to be about 627 nm, indicating that PM6 suffered severe entanglement in their solutions (see *Joule*, **2023**, 7, 2386-2401). In addition, the crystallization of PM6 in the PM6:Y6 and PM6:L8-BO solutions was determined significantly faster than that of the acceptors. Consequently, the PM6:Y6 and PM6:L8-BO blend films could easily form a large domain size and lead to the unfavorable phase separation and morphology. In terms of the PM6:BTA-E3 blend films, the absorption peak of PM6 and BTA-E3 changed at the same time, indicating that the donor and acceptor crystallized simultaneously, while the donor crystallization occurs before the acceptor during the PM6:BTA-C6 blend films evolution, demonstrated by the absorption peak change of PM6 came before that of BTA-C6. Due to the introduction of H-bonding, the PM6:BTA-E3 blend films demonstrated increased homogeneous and heterogeneous interactions, and consequently exhibited the rapid and synchronous crystallization (see *Adv. Mater.* **2024**, 36, 2305356). As a result, the PM6:BTA-E3 blend films could obtain the proper phase separation and optimal domain size along with the fiber-like bicontinuous network, thereby contributing to the improved device performance.”.

Supplementary Fig. 53. a-d, 2D *in situ* UV-vis absorption of PM6:Y6 (a), PM6:L8-BO

(b), PM6:BTA-C6 (c) and PM6:BTA-E3 (d) blend films during film evolution. e-h, Time-dependent absorption peak shifts during film formation for the corresponding acceptors and PM6.

2. How about the trend of photovoltaic properties among BTA-C6, BTA-E3, BTA-E6, and BTA-E when the device is prepared using halogenated solvents? Please, explain the results and not just describe the result. It is necessary to explain the relationship between chemical structures of the acceptors and properties of the solvents.

Response: Thanks for the reviewer's suggestion. In this work, we had tried different solvent to fabricate the OSCs for characterization of photovoltaic performance, including the halogenated solvent chloroform (CF) and non-halogenated solvent *o*-xylene (*o*-XY). Notably, the photovoltaic performance of the BTA-series acceptors was found to be excellent both in the *o*-XY and CF processed devices. In consideration of environmental issues, we selected the environmentally friendly green solvent *o*-XY as the processing solvent.

We added the corresponding discussion in Supplementary Note I of Supporting Information: “**Supplementary Note I**

Supplementary Note I

The CF processed devices were also fabricated with the conventional structure of ITO/PEDOT:PSS/ active layers/PDINN/Ag. The *J-V* curves of the optimized CF processed OSCs are shown in Supplementary Fig. 52, and the corresponding photovoltaic performance parameters of the CF processed devices are summarized in Supplementary Table 7. When processed with CF, all the four acceptors exhibited almost identical PCE with that of the *o*-XY processed devices, with the corresponding PCE of 17.22%, 18.24%, 16.13% and 14.72% for the OSCs based on PM6:BTA-C6, PM6:BTA-E3, PM6:BTA-E6 and PM6:BTA-E9 respectively. Taking into account environmental concerns, we have opted for the eco-friendly green solvent *o*-XY as the processing solvent.”.

Meanwhile, we modified the sentence “It needs to be emphasized that environmentally friendly green solvent *o*-xylene was employed as the processing solvent” on page 9 to “It needs to be emphasized that environmentally friendly green solvent *o*-xylene was employed as the processing solvent (Supplementary Note I and Note II)”.

Supplementary Fig. 52. *J-V* curves of the CF processed OSCs under the illumination of AM 1.5 G, 100 mW cm⁻².

Supplementary Table 7. Photovoltaic performance parameters of the CF processed OSCs under the illumination of AM 1.5 G, 100 mW cm⁻².

Active Layer	V_{oc} (V)	J_{sc} (mA cm ⁻²)	FF (%)	PCE (%)
PM6:BTA-C6	0.837	26.82	76.73	17.22
PM6:BTA-E3	0.836	28.05	77.79	18.24
PM6:BTA-E6	0.829	26.64	73.01	16.13
PM6:BTA-E9	0.816	25.12	71.80	14.72

3. It is already known that non-covalent bonding such as hydrogen bonding is favorable for mechanical stability of the BHJ film. How about the stability of the BHJ film under high humidity? It is possible that the hydrogen bonding of the acceptor makes the BHJ more hydrophilic and allows the water molecules to easily diffuse inside.

Response: Thanks for the reviewer's suggestion. Firstly, although the presence of H-bonding effectively improves the morphology and mechanical stability of the blend film, H-bonding interactions still only account for a small part of the molecular interactions of the entire blend film, and thus have little effect on the hydrophilicity of the BHJ film.

Secondly, to address the device stability issue when exposed to the water, device encapsulation is a common and effective strategy (*Aggregate*. **2024**, <https://doi.org/10.1002/agt2.567>). A high-quality encapsulation can effectively prevent the moisture penetration, thereby protecting the internal structure of OSCs and prolonging the stability of OSC lifetime. Under the robust encapsulation condition, the stability problems caused by the highly hydrophilic PEDOT:PSS can also be solved (*Adv. Funct. Mater.* **2023**, *33*, 2305445). Therefore, the change in hydrophilicity of BHJ film caused by H-bonding has minimal impact on device stability.

Response to Referee #3:

This manuscript is about introducing hydrogen bonding to enhance stretchability and achieve remarkable efficiencies of OPVs. However, the concept of this study is not sufficiently novel to be published in “Nature Communications”. This approach is well known in the organic semiconductor community (e.g. Chem. Mater. 2020, 32, 13, 5700–5714, Adv. Mater. 2022, 34, 2207544, Chem. Mater. 2023, 35, 24, 10476–10486). Additionally, the crack onset strain of active layer system with polymer donor and small molecule acceptor have been improved to over 13% (e.g. J. Am. Chem. Soc. 2023, 145, 11914–11920), almost doubling the COS value reported in this paper. Furthermore, while the authors claim mechanical robustness for organic solar cells, they lack measurements for the mechanical durability of flexible/stretchable OPVs. Therefore, I recommend submitting this manuscript to more specialized journals rather than “Nature Communications”. I also include comments to improve this manuscript.

Response: As for the novelty of this work, we give the following explanations:

(1) Introducing H-bonding into organic semiconductors is an effective strategy for achieving high performance and mechanically robust OSCs. The recent works mentioned by the reviewer are mainly dedicated to incorporate H-bonding into the polymer materials through the ternary copolymerization method, including donors and acceptors, such as PhAm5 (*Adv. Mater.* **2022**, 34, 2207544), PM7-Thy10 (*J. Am. Chem. Soc.* **2023**, 145, 11914–11920), and N2200-ThyDap (*Chem. Mater.* **2023**, 35, 10476-10486). However, the random ternary copolymerization presents challenges for the repeatability of synthesis as well as the reliability of device photovoltaic performance and mechanical robustness. Moreover, since the groups that can form H-bonding are usually introduced by the third component in the ternary copolymerization, therefore, the H-bonding introduced by this method are very limited. In order to solve the problems of repeatability, reliability and effectiveness of introducing H-bonding in the active layer of OSCs, **in this work, we firstly attempt to introduce the dynamic chemical bonds (H-bonding) into the A-DA'D-A type small molecule acceptors, by selecting ethyl ester group as the unit providing H-bonding interactions.** This has important implications for introducing dynamic chemical bonds into the design of molecules that can be used in OSCs.

(2) We also noticed that the COS values of other work have been improved to over 13% (*Adv. Mater.* **2022**, 34, 2207544, and *J. Am. Chem. Soc.* **2023**, 145, 11914–11920). However, there is a lack of the general and standardized guidelines for the film-on-water (FOW) measurement and the detailed performing methods are various from each research group, such as film processing method, film shape, adhesion force and strain rate. Therefore, we hold the opinion that the COS values are not suitable for comparison across the different works due to these variations in measurement techniques. For example, the COS value of the PM6:BTP-eC9 blend film was determined to be 2.99% (*Adv. Energy Mater.* **2022**, 12, 2202224) and 5.5% (*Adv. Mater.* **2023**, 36, 2309379) in different research groups. **In this work, the blend films of PM6:BTA-E3 (7.0%), PM6:BTA-E6 (9.3%) and PM6:BTA-E9 (8.3%) exhibit the improved COS values compared to that of their counterpart blend film of PM6:BTA-C6 (4.3%), indicating that the modification of ethyl ester side chain could effectively enhance the stretchability and toughness of the active layer films.**

(3) In order to investigate the impact of dynamic H-bonding on the photovoltaic performance and mechanical robustness of OSCs, we fabricated the flexible devices with the structure of PET/ITO/2PACz/Active layer/PDINN/Ag. The current density-voltage curves ($J-V$) of the flexible OSCs are shown in **Supplementary Fig. 50a** and the corresponding photovoltaic performance parameters are listed in **Supplementary Table 6**. **Notably, the PM6:BTA-E3 based flexible devices exhibited an outstanding PCE of 18.33% with the V_{oc} of 0.829 V, J_{sc} of 28.18 mA cm⁻², and FF of 78.38%. While, the PM6:BTA-C6 based counterparts only achieved a PCE of 17.18%. Moreover, the mechanical durability of the flexible OSCs was tested at a bending radius of 5 nm. As depicted in **Supplementary Fig. 50b**, the PM6:BTA-E3 based flexible devices maintained 88.91% of its initial PCE after 2000 consecutive bends, while the PM6:BTA-C6 based devices rapidly decayed to 77.85%. The superior device performance and mechanical endurance of the PM6:BTA-E3 based flexible devices imply that the dynamic H-bonding modification could effectively and simultaneously improve the performance and durability of the flexible OSCs.**

We added the corresponding description on Page 22: “**In order to investigate the impact of dynamic H-bonding on the photovoltaic performance and mechanical robustness**

of OSCs, we fabricated the flexible devices with the structure of PET/ITO/2PACz/Active layer/PDINN/Ag. Considering that BTA-E3 has the highest device efficiency and a similar molecular structure to BTA-C6, we selected BTA-E3 and BTA-C6 as the acceptor materials of active layer to more intuitively reflect the effect of H-bonding modification on the mechanical durability of the flexible OSCs. The J - V curves of the flexible OSCs are shown in **Supplementary Fig. 50a** and the corresponding photovoltaic performance parameters are listed in **Supplementary Table 6**. Notably, the PM6:BTA-E3 based devices exhibited an outstanding PCE of 18.33% with the V_{oc} of 0.829 V, J_{sc} of 28.18 mA cm^{-2} , and FF of 78.38%. While the PM6:BTA-C6 based devices only achieved a PCE of 17.18%. Moreover, the mechanical durability of these flexible OSCs was tested at a bending radius of 5 nm. As depicted in **Supplementary Fig. 50b**, the PM6:BTA-E3 based flexible devices maintained 88.91% of its initial PCE after 2000 consecutive bends, while the PCE of the PM6:BTA-C6 based devices decayed to 77.85% of its initial PCE under the same bending condition. The superior device performance and mechanical endurance of the PM6:BTA-E3 based flexible devices imply that the dynamic H-bonding modification could effectively and simultaneously improve the performance and durability of the flexible OSCs.” And we added the sentence: “The PM6:BTA-E3 based flexible OSCs also exhibit enhanced mechanical durability than that of the PM6:BTA-C6 based devices.” on Page 25.

Supplementary Fig. 50. a, J - V curves of the flexible OSCs under the illumination of AM 1.5 G, 100 mW cm^{-2} . **b**, PCE retention of the flexible devices as a function of bending cycle number during the bending cyclic test. **c**, Photograph of the flexible OSC.

Supplementary Table 6. Photovoltaic performance parameters of the flexible OSCs under the illumination of AM 1.5 G, 100 mW cm^{-2} .

Active Layer	V_{oc} (V)	J_{sc} (mA cm ⁻²)	FF (%)	PCE (%)	PCE retention ^a (%)
PM6:BTA-C6	0.828	27.28	76.02	17.18	77.85
PM6:BTA-E3	0.829	28.18	78.38	18.33	88.91

^a The PCE retention after 2000 bending circles.

1. I am curious about the hydrogen bonding C-H...F. Based on the chemical structure, the other materials demonstrate the same H bonding units, could we observe the H bonding in the other materials? Please discuss this in detail.

Response: Thanks for the reviewer's suggestion. The H-bonding between molecules of BTA-E3 was directly observed through its single-crystal XRD measurement. However, obtaining single-crystal data for BTA-E6 and BTA-E9 are challenging due to the extended alkyl chain. In addition, the presence of severe twin crystal also poses significant challenges for single-crystal analysis. Therefore, we couldn't directly observe the H-bonding in the other materials, but we could certify the existence of H-bonding from other circumstantial evidence.

As shown in Fig. 1e, the C-H...F H-bonding of BTA-E3 is generated between the F atom of the terminal group and the H atom of methylene adjacent to the carbonyl group in another dimer. It is reasonable to assume that C-H...F H-bonding may also exist in BTA-E6 and BTA-E9 due to the presence of the same ethyl ester motif in their structure. Furthermore, based on measurements of mechanical properties and thermal stability, it can be observed that PM6:BTA-E6 and PM6:BTA-E9 blend films exhibited increased mechanical performance and improved thermal stability compared to their PM6:BTA-C6 counterparts. These improvements can be attributed to the presence of H-bonding within these materials.

To clarify the effect of H-bonding on BTA-E6 and BTA-E9, we added the corresponding description "In addition, as the ethyl ester side chain length of the SMAs elongating, the blend films of PM6:BTA-E6 and PM6:BTA-E9 exhibited gradually increased toughness of 0.83 MJ m⁻³ and 0.94 MJ m⁻³ respectively (Supplementary Fig. 49). The enhancement in toughness unequivocally signifies an enhanced capacity to withstand

crack and deformation. In a nutshell, the dynamic H-bonding undeniably harbors immense potential in augmenting stretchability and toughness of the films.” on Page 22 and the sentence: “The PCE decay of the PM6:BTA-E6 and PM6:BTA-E9 based OSCs under thermal stress was also enhanced with the respective PCE retention of 95.4% and 93.7% after annealing (Supplementary Fig. 51a)” on Page 23.

2. *How about the H-bonding strength in BTA-E3?*

Response: Thanks for the reviewer’s suggestion. According to the single-crystal data, the bonding energy of the H-bonding in BTA-E3 was calculated to be 3.585 kJ mol⁻¹.

3. *How do we understand the effect of H-bonding strength on devices’ thermal stability? H-bonding is thermodynamically reversible, is the thermal stability also reversible?*

Response: Thanks for the reviewer’s comments. Firstly, H-bonding can promote the formation of non-covalent cross-linked networks between molecules (Fig. 6d) in the active layer, which inhibit the diffusion of photovoltaic materials and improve the morphological stability of the blend film. Secondly, H-bonding is thermodynamically reversible and there is a spontaneously dynamic equilibrium of association/disassociation within H-bonding (*Polymer*, **2020**, 203, 122787). When subjected to external stimuli (such as thermal stress), the dynamic H-bonding can be disrupted, enabling efficient dissipation of energy, meanwhile the new H-bonding will formed and bestowing materials with excellent thermal tolerance.

We added the corresponding description on the Page 23: “The results imply that incorporating the dynamic H-bonding intermolecular interaction into the molecular design may be a very effective strategy to promote the formation of non-covalent cross-linked networks between photovoltaic materials (Fig. 6d) and significantly improve both the resistance against the thermal stress dissipation and the morphological stability.”.

4. *The authors mentioned that the 2PACz HTL is better than PEDOT:PSS, leading to the new record PCE. The detailed discussions of the PCE enhancement are lacking.*

Response: Thanks for the reviewer’s suggestion. The improved PCE of 2PACz-based devices could be attributed to the reduced parasitic absorption of hole transport layer (*Adv.*

Mater. **2024**, *36*, 2400342) and more favorable vertical phase separation of active layer of devices. Consequently, when replacing PEDOT:PSS with 2PACz, the PM6:BTA-E3 based devices achieved a superior PCE of 19.92%.

To clarify the PCE enhancement of the devices using 2PACz as HTL, we added the sentence: “The improved PCE of the OSCs with the 2PACz hole transport layer (HTL) could be attributed to the reduced parasitic absorption of the HTL and the more favorable vertical phase separation of active layer in the devices, which will be discussed later.” on Page 10 and discussed the PCE enhancement of the devices using 2PACz as HTL in the FLAS measurement part at Page 19: “In addition, to illustrate PCE enhancement of the 2PACz-based OSCs, the vertical phase distribution of the PM6:BTA-E3 blend films were investigated when different HTLs were employed. Notably, a favored vertical phase distribution was observed in the 2PACz-based devices, wherein the donors were enriched at the lowermost regions of the film close to the HTL (Supplementary Fig. 47). Besides, the calculated G values of the PM6:BTA-E3 blend film on the 2PACz substrate were higher than that on the PEDOT:PSS HTL. The more favorable vertical phase separation and more excitons generated could account for the enhanced J_{sc} and FF of the 2PACz based OSCs.”.

Supplementary Figure 47. a,d, Components distribution profiles of the PM6:BTA-E3 blend films on (a) PEDOT:PSS and (d) 2PACz substrate at different film-depths. b,e, Dependence of the simulated exciton generation rate (G) on the film depth of PM6:BTA-E3 blend films on (b) PEDOT:PSS and (e) 2PACz substrate. c,f, Numerical simulations for the exciton generation contours of PM6:BTA-E3 blend films on (c) PEDOT:PSS and (f)

2PACz substrate.

5. The authors claimed that this OPV shows a new recorded PCE in binary devices processed with non-halogenated solvents. The authors should provide references to compare the efficiency.

Response: Thanks for the reviewer's suggestion. The photovoltaic performance of the OSCs processed by non-halogenated solvents was summarized in the Supplementary Table 1. As depicted in Supplementary Table 1, the PCE of 19.92% achieved by the PM6:BTAE3 based devices was the highest PCE value in non-halogenated solvents processed OSCs. We added the sentence: "This PCE stands as the highest recorded PCE in the devices processed by green solvents (Supplementary Table 1)." on Page 10 and the Supplementary Table 1 was added into the Supporting Information.

Supplementary Table 1. Summary of photovoltaic performance parameters of the non-halogenated solvent processed OSCs with the PCEs over 17%.

Active Layer	Solvent	V_{oc} (V)	J_{sc} (mA cm ⁻²)	FF (%)	PCE (%)	Ref
PM6:Y6:Y-4C-4O	o -XY	0.87	25.42	77.0	17.03	1
PM6:PTQ10:BTP-eC9	o -XY	0.855	27.86	80.2	19.10	2
D18:L8-BO	CS ₂ /PX	0.885	26.25	75.30	17.50	3
PM6:CH8-4	o -XY	0.900	25.51	75.1	17.27	4
PM6:G-Trimer	o -XY	0.896	26.75	79.30	19.01	5
PBQ6:PYF-T- o	Tol	0.886	25.12	76.64	17.06	6
PM6:BTP-eC9	Tol	0.865	27.43	76.55	18.16	7
PM6:Y6:BTO:PC ₇₁ BM	PX	0.85	27.12	75.75	17.41	8
D18-Cl:L8-BO-X	Tol	0.893	26.78	79.6	19.04	9
PM6: BTP-eC9	o -XY	0.846	28.2	78.8	18.8	10
PM6:CB16	o -XY	0.92	25.98	76.89	18.32	11
PM6:BTP-eC9	o -XY	0.851	27.65	79.0	18.6	12
PM6:BTP-eC9:PY-IT	o -XY	0.859	27.79	81.3	19.41	13
PM6:BTP-BO-4F:GS-ISO	o -XY	0.865	27.55	78.19	18.63	14
PM6:L8-BO:BTO-BO	Tol	0.881	27.14	81.23	19.42	15

PM6:D18-Cl:L8-BO	Tol/CS ₂	0.872	27.29	79.10	18.83	16
PM6:L8-BO-X:Tri-V	o -XY	0.890	27.56	80.8	19.82	17
PDTP-BDD:D18:L8-BO	o -XY	0.905	26.90	79.53	19.36	18
D18:L8-BO:PY-TPT	o -XY	0.870	25.39	78.96	17.45	19
PL1:BTP-eC9-4F	o -XY	0.876	27.11	76.41	18.14	20
PM6:BTP-eC11:BN-T	o -XY	0.856	26.54	79.3	18.02	21
PM6:EV-i	o -XY	0.897	26.60	76.56	18.27	22
PM6:L8-Ph	o -XY	0.870	26.40	80.11	18.40	23
PM6:BTP-eC9	o -XY	0.847	27.22	80.31	18.52	24
D18:DTC11	o -XY/CS ₂	0.858	27.5	80.5	19.0	25
PM6:L15	o -XY	0.93	25.95	77.26	18.72	26

6. Please specify the statistic number when demonstrated the error bars in the figures.

Response: Thanks for the reviewer's suggestion. We have added the statistic number in Fig. 2: "e, Hole and electron mobilities of the corresponding OSCs. **Error bars represent the standard error of the mean (n = 5).**", and in Fig. 6: "c, Normalized PCE of the OSCs based on PM6:BTA-C6 and PM6:BTA-E3 during annealing at 100 °C for 250 h. **Error bars represent the standard error of the mean (n = 10).**".

7. Please specify the full name of PM7.

Response: Thanks for the reviewer's suggestion. We have added the corresponding description on Page 4: "PM7 (Poly[(2,6-(4,8-bis(5-(2-ethylhexyl-3-chloro)thiophen-2-yl)-benzo[1,2-b:4,5-b]dithiophene))-alt-(5,5-(1,3-di-2-thienyl-5,7-bis(2-ethylhexyl)benzo[1,2-c:4,5-c]dithiophene-4,8-dione))]"

Supplementary References

1. Kim, C. *et al.* Impact of the molecular structure of oligo(ethylene glycol)-incorporated Y-series acceptors on the formation of alloy-like acceptors and performance of non-halogenated solvent-processable organic solar cells. *ACS Appl. Mater. Interfaces* **15**, 24670–24680 (2023).
2. Ma, R. *et al.* Revealing the underlying solvent effect on film morphology in high-efficiency organic solar cells through combined *ex situ* and *in situ* observations. *Energy Environ. Sci.* **16**, 2316–2326 (2023).
3. Su, Y. *et al.* High-efficiency organic solar cells processed from a halogen-free solvent system. *Sci. China Chem.* **66**, 2380–2388 (2023).
4. Chen, H. *et al.* Terminally chlorinated and thiophene-linked acceptor-donor-acceptor structured 3D acceptors with versatile processability for high-efficiency organic solar cells. *Angew. Chem. Int. Ed.* **62**, e202307962 (2023).
5. Wang, C. *et al.* Unique assembly of giant star-shaped trimer enables non-halogen solvent-fabricated, thermal stable, and efficient organic solar cells. *Joule* **7**, 2386–2401 (2023).
6. Hu, K. *et al.* Solid additive tuning of polymer blend morphology enables non-halogenated-solvent all-polymer solar cells with an efficiency of over 17%. *Energy Environ. Sci.* **15**, 4157–4166 (2022).
7. Zhang, J. *et al.* Polymer-entangled spontaneous pseudo planar heterojunction for constructing efficient flexible organic solar cells. *Adv. Mater.* **36**, 2309379 (2023).
8. Chen, H. *et al.* A guest-assisted molecular-organization approach for >17% efficiency organic solar cells using environmentally friendly solvents. *Nat. Energy* **6**, 1045–1053 (2021).
9. Luo, S. *et al.* Auxiliary sequential deposition enables 19%-efficiency organic solar cells processed from halogen-free solvents. *Nat. Commun.* **14**, 6964 (2023).
10. He, W. *et al.* In situ self-assembly of trichlorobenzoic acid enabling organic photovoltaics with approaching 19% efficiency. *Adv. Funct. Mater.* **34**, 2313594 (2023).
11. Xue, Y.-J. *et al.* Unraveling the structure–property–performance relationships of fused-ring nonfullerene acceptors: toward a C-shaped *ortho* -benzodipyrrole-based

- acceptor for highly efficient organic photovoltaics. *J. Am. Chem. Soc.* **146**, 833–848 (2023).
12. Xu, L. *et al.* Volatile solid-assisted molecular assembly enables eco-friendly processed organic photovoltaic cells with high efficiency and photostability. *Adv. Funct. Mater.* **34**, 2314178 (2024).
 13. Zhang, Y. *et al.* Achieving 19.4% organic solar cell via an *in situ* formation of p-i-n structure with built-in interpenetrating network. *Joule* **8**, 509–526 (2024).
 14. Huang, T. *et al.* 18.63% efficiency of ternary organic solar cells achieved via nonhalogenated solvent and hot spin-coating process. *Adv. Funct. Mater.* 2315825 (2024) doi:10.1002/adfm.202315825.
 15. Chen, H. *et al.* Heterogeneous nucleating agent for high-boiling-point nonhalogenated solvent-processed organic solar cells and modules. *Adv. Mater.* 2402350 (2024) doi:10.1002/adma.202402350.
 16. Zhang, Z. *et al.* Synchronous regulation of donor and acceptor microstructure using thiophene-derived non-halogenated solvent additives for efficient and stable organic solar cells. *Adv. Funct. Mater.* 2401823 (2024) doi:10.1002/adfm.202401823.
 17. Song, J. *et al.* Non-halogenated solvent-processed organic solar cells with approaching 20 % efficiency and improved photostability. *Angew. Chem. Int. Ed.* **63**, e202404297 (2024).
 18. He, Y. *et al.* Developing benzodithiophene-free donor polymer for 19.36% efficiency green-solvent-processable organic solar cells. *Chem. Eng. J.* **490**, 151920 (2024).
 19. Wei, Y. *et al.* High performance as-cast organic solar cells enabled by a refined double-fibril network morphology and improved dielectric constant of active layer. *Adv. Mater.* 2403294 (2024) doi:10.1002/adma.202403294.
 20. Lu, H. *et al.* Random terpolymer enabling high-efficiency organic solar cells processed by nonhalogenated solvent with a low nonradiative energy loss. *Adv. Funct. Mater.* **32**, 2203193 (2022).
 21. Ma, R. *et al.* *In situ* and *ex situ* investigations on ternary strategy and co-solvent effects towards high-efficiency organic solar cells. *Energy Environ. Sci.* **15**, 2479–2488 (2022).
 22. Zhuo, H. *et al.* Giant molecule acceptor enables highly efficient organic solar cells

- processed using non-halogenated solvent. *Angew. Chem. Int. Ed.* **62**, e202303551 (2023).
23. Wu, X. *et al.* Introducing phenyl end group in the inner side chains of a-da'd-a acceptors enables high-efficiency organic solar cells processed with nonhalogenated solvent. *Adv. Mater.* **35**, 2302946 (2023).
 24. Yang, C. *et al.* Hot-casting strategy empowers high-boiling solvent-processed organic solar cells with over 18.5% efficiency. *Adv. Mater.* **36**, 2305356 (2024).
 25. Zhong, Z. *et al.* Non-halogen solvent processed binary organic solar cells with efficiency of 19% and module efficiency over 15% enabled by asymmetric alkyl chain engineering. *Adv. Energy Mater.* **13**, 2302273 (2023).
 26. Liu, B. *et al.* Isomerized green solid additive engineering for thermally stable and eco-friendly all-polymer solar cells with approaching 19% efficiency. *Adv. Mater.* **35**, 2308334 (2023).

REVIEWER COMMENTS

Reviewer #1 (Remarks to the Author):

The revised version represents a substantially revised effort in response to the reviewers' concerns. Important control experiments have been added and enhance support for the general conclusions claimed. This version is suitable for publication.

Reviewer #2 (Remarks to the Author):

Although some questions are answered, there are still critical issues to be addressed.

- Regarding the stability of the BHJ film under high humidity, the authors need to carefully compare the stability of the devices prepared with BTA-C6 or BTA-E3.

- Regarding the question of "What are the effects of the hydrogen-bonding to make nonhalogenated solvent-based process favorable?"

The authors mentioned that "Due to the introduction of H-bonding, the PM6:BTA-E3 blend films demonstrated increased homogeneous and heterogeneous interactions, and consequently exhibited the rapid and synchronous crystallization. As a result, the PM6:BTA-E3 blend films could obtain the proper phase separation and optimal domain size along with the fiber-like bicontinuous network." This explanation is vague and based on rather incoherent assumption.

Reviewer #3 (Remarks to the Author):

Based on the authors' responses, they have comprehensively answered the reviewer's questions and made corresponding changes and additions to the manuscript. Although the revised manuscript has improved in quality, there are still critical aspects that need attention. It would be beneficial if the authors could clarify the following issues.

1. Please provide more detailed evidence to substantiate the novelty of this work and include sufficient explanations in the main manuscript to help readers understand how the current work compares to the benchmark.

For example: (1) The limitations of ternary copolymerization, such as the repeatability of synthesis, device reliability, and mechanical robustness, were not mentioned in reference 39. How did you reach these conclusions? (2) The paper (Chem. Mater. 2020, 32, 13, 5700–5714) includes a comprehensive study on the introduction of hydrogen bonding into materials. What are the remaining challenges in incorporating hydrogen bonding into your materials compared to previous work, especially when you state this is the first attempt?

2. Please explain how the COS increases from 4.3% to 7.0% are accompanied by a 50 MPa decrease in elastic moduli for PM6:BTA-C6 and PM6:BTA-E3, whereas the COS increases from 7.0% to 9.6% are accompanied by a 200 MPa decrease in elastic moduli for PM6:BTA-E3 and PM6:BTA-E6.

3. The authors attribute both the improved COS values and thermal stability to the introduction of hydrogen bonding. However, while PM6:BTA-E6 demonstrates the best COS value of 9.6%, PM6:BTA-E3 devices show the best thermal stability after introducing hydrogen bonding. Could you clarify the reason for this discrepancy?

Response letter to Reviewers

Response to Reviewer #1:

The revised version represents a substantially revised effort in response to the reviewers' concerns. Important control experiments have been added and enhance support for the general conclusions claimed. This version is suitable for publication.

Response: We thank the reviewer's positive comments!

Response to Reviewer #2

Although some questions are answered, there are still critical issues to be addressed.

1. *Regarding the stability of the BHJ film under high humidity, the authors need to carefully compare the stability of the devices prepared with BTA-C6 or BTA-E3.*

Response: Thanks for the reviewer's suggestion! Our response are as follows:

(1) We fabricated the OSCs based on PM6:BTA-E3 and PM6:BTA-C6 with the device structure of ITO/PEDOT:PSS/active layers/PDINN/Ag, and investigated the stability of the corresponding devices under the 50% relative humidity condition in a glovebox (Figure R1a). The corresponding encapsulated devices based on PM6:BTA-E3 and PM6:BTA-C6 exhibited excellent stability with over 94% PCE retention after 144 hours exposing to moisture (Figure R1b). **This result indicates the BTA-E3-based device has high stability and the H-bonding of the acceptors has minimal impact on encapsulated device stability even under high humidity condition (Supplementary Figs. 51c-d).** (see line 5 from bottom in p. 23.)

(2) To further investigate whether the H-bonding of the acceptors make the active layer more hydrophilic and thus affect the stability of device, the un-encapsulated devices based on both the PM6:BTA-E3 and PM6:BTA-C6 were tested under the same humidity condition. As shown in Figure R1b, the performance of both devices decayed rapidly when exposed to moisture and the PCE of these OSCs fast decreased to 30% of their initial PCE after 8 hours aging. Specifically, despite introducing the H-bonding into the acceptors, the PM6:BTA-E3 based un-encapsulated devices exhibited slight better stability against the moisture than those of the device based on PM6:BTA-C6, which indicates that the **H-bonding primarily played a role of preservation of the blend film morphology rather than promoting the moisture penetration.**

(3) The fast decreased PCE of the un-encapsulated devices are mainly come from the highly hydrophilic PEDOT:PSS. To illustrate this problem, we also fabricated the PEDOT:PSS-only devices with the structure of ITO/PEDOT:PSS/Ag to explore the influence of PEDOT:PSS to the OSCs in the humidity stability test. The film conductivity of PEDOT:PSS-only devices was calculated with the formula: $\sigma = \frac{L}{RS}$, where the L is the film thickness, R is the resistance, S is the film area. As shown in Figure R1c, **the decay tendency of the film conductivity was consistent**

with the PCE decay of the corresponding OSCs and the PEDOT:PSS-only devices barely preserved the 50% conductivity of the initial devices when exposed to moisture for 8 hours.

Overall, the change in hydrophilicity of BHJ film caused by H-bonding has little effect on device stability. The encapsulated BTA-E3-based device exhibited high stability even under high humidity condition.

In order to elucidate the stability of the BHJ film under high humidity condition, we added the corresponding discussion on Page 23: “In addition, the PM6:BTA-E3 based OSCs also exhibited remarkable long-term stability storing in the nitrogen glove box (Supplementary Fig. 51b), and even under high humidity condition (Supplementary Figs. 51c-d).” and revised the corresponding Supporting Fig. 51 in the Supplementary Information.

Figure R1. **a**, Photo of the stability test under 50% relative humidity condition. **b**, PCE retention of the encapsulated and unencapsulated devices of the PM6:BTA-E3 and PM6:BTA-C6 during exposing to 50% relative humidity condition for 144 hours. **c**, Film conductivity retention of the PEDOT:PSS-only devices during exposing to 50% relative humidity condition for 144 hours. Error bars represent the standard error of the mean ($n = 5$).

2. Regarding the question of “What are the effects of the hydrogen-bonding to make nonhalogenated solvent-based process favorable?”

The authors mentioned that “Due to the introduction of H-bonding, the PM6:BTA-E3 blend films demonstrated increased homogeneous and heterogeneous interactions, and consequently exhibited the rapid and synchronous crystallization. As a result, the PM6:BTA-E3 blend films could obtain the proper phase separation and optimal domain size along with the fiber-like bicontinuous network.” This explanation is vague and based on rather incoherent assumption.

Response: Since the experimental results used to illustrate the effects of the H-bonding on non-

halogenated solvent processing are scattered in the main text and SI respectively, we apologize for this confusion caused to the reviewer. To better understand why the introduction of H-bonding can make the BTA-E3 acceptor more suitable for non-halogenated solvent processing than BTA-C6, we further correlated the experimental results.

Firstly, based on the observation in *GIWAXS* measurement of neat and blend films (Supplementary Fig. 44 and Fig. 4), it can be concluded that due to the introduction of H-bonding, **the homogeneous interactions of BTA-E3 were enhanced**, resulting in the more **ordered and dense π - π stacking** of BTA-E3 than that of BTA-C6 in the film. In addition, compared with BTA-C6, the Flory-Huggins interaction parameters (χ) (Supplementary Fig. 48) exhibits that BTA-E3 possesses **increased heterogeneous interactions with PM6**, as evidenced by the smaller χ value, which will **improve the miscibility of BTA-E3 and PM6**. Such enhanced homogeneous and heterogeneous interactions might lead to different assembly behavior during the film evolution.

The *in situ UV-vis absorption spectra* of the blend films processed with *o*-xylene were monitored to elucidate the evolution of the blend film during the coating process (Supplementary Fig. 53). Due to the absorption peak location could reflect the aggregation properties of molecules, the time-dependent maximum absorption peak shift curves of the donor and acceptor were extracted to investigate the individual aggregation behavior of the donor and acceptor. As shown in Supplementary Fig. 53g, the peak location of PM6 in the PM6:BTA-C6 blend film started to show the gradually redshift at 7.7 s, while the absorption peak of the BTA-C6 barely changed at the same time, which suggested that donor aggregation came before the acceptor during the film evolution. While in terms of the PM6:BTA-E3 blend film (Supplementary Fig. 53h), the absorption peak of PM6 and BTA-E3 almost changed at the same time (from 8.4 s to 10.7 s), indicating that the donor and acceptor crystallized simultaneously (*Adv. Mater.* **2024**, 36, 2305356). Besides, the PM6:BTA-E3 blend films exhibited the shorter film formation duration (c.a. 2.3 s) than that of PM6:BTA-C6 counterpart (c.a. 2.5 s), which mainly resulted from the enhanced homogeneous and heterogeneous interactions. **The long assembly process might lead to the excessive aggregation and large domain size as evidenced by the AFM and TEM image (Fig. 5)**. As a result, benefitted from the introduction of H-bonding, **the PM6:BTA-E3 blend film exhibited a rapid and synchronous film evolution process, resulting in optimal phase separation along with the fiber-like bicontinuous network (Fig. 5)**, which is conducive to charge generation and extraction and contributing to the improved device performance.

In summary, the introduction of H-bonding enhanced the homogeneous and heterogeneous interactions, and such difference in the interactions caused different assembly behavior during the film evolution. We further investigated how these interactions led to variations in film formation processes by *in situ* UV-vis absorption spectra test, and we observed that the BTA-E3-based blend films exhibited more ordered molecular stacking and proper phase separation. These evidences support the conclusion that, compared to BTA-C6, BTA-E3 is more conducive to processing with non-halogenated solvents due to the presence of H-bonding.

To further clarify this point, we revised the **Supplementary Note II** in the Supplementary Information in pp. S50-S51: “Compared with BTA-C6, the introduction of H-bonding enhanced the homogeneous and heterogeneous interactions of BTA-E3 (Fig. 4 and Supplementary Fig. 48), and such difference in the interactions caused different assembly behavior during the film evolution. The *in situ* UV-vis absorption spectra test shows how these interactions led to differences in film formation processes of BTA-C6 and BTA-E3. In terms of the PM6:BTA-C6 blend films (Supplementary Fig. 53g), the peak location of PM6 absorption in the PM6:BTA-C6 blend film started to show the gradually redshift at 7.7 s, while the absorption peak of BTA-C6 barely changed at the same time, which suggested that donor aggregation came before the acceptor during the film evolution. While for the PM6:BTA-E3 blend film (Supplementary Fig. 53h), the absorption peak of PM6 and BTA-E3 almost changed at the same time (from 8.4 s to 10.7 s), indicating that the donor and acceptor crystallized simultaneously. Besides, the PM6:BTA-E3 blend films exhibited the shorter film formation duration (c.a. 2.3 s) than that of PM6:BTA-C6 counterpart (c.a. 2.5 s), which mainly resulted from the enhanced homogeneous and heterogeneous interactions. The long film formation process might lead to the excessive aggregation and large domain size as evidenced by the AFM and TEM images (Fig. 5). As a result, benefitted from the introduction of H-bonding, the PM6:BTA-E3 blend film exhibited a rapid and synchronous film evolution process, resulting in optimal phase separation along with the fiber-like bicontinuous network (Fig. 5), which is conducive to charge generation and extraction and contributing to the improved device performance.”

Response to Reviewer #3

Based on the authors' responses, they have comprehensively answered the reviewer's questions and made corresponding changes and additions to the manuscript. Although the revised manuscript has improved in quality, there are still critical aspects that need attention. It would be beneficial if the authors could clarify the following issues.

1. Please provide more detailed evidence to substantiate the novelty of this work and include sufficient explanations in the main manuscript to help readers understand how the current work compares to the benchmark.

For example: (1) The limitations of ternary copolymerization, such as the repeatability of synthesis, device reliability, and mechanical robustness, were not mentioned in reference 39. How did you reach these conclusions? (2) The paper (Chem. Mater. 2020, 32, 13, 5700–5714) includes a comprehensive study on the introduction of hydrogen bonding into materials. What are the remaining challenges in incorporating hydrogen bonding into your materials compared to previous work, especially when you state this is the first attempt?

Response: Thanks for the reviewer's suggestions. These questions the reviewer raised provide a good perspective to illustrate the novelty of this work. We will elaborate on the uniqueness of our work through these two focused concerns.

(1) For the limitations of ternary copolymerization.

It is well known that the polymers are mixtures, and most conjugated polymers used in organic semiconductors usually suffer from batch-to-batch variation due to the differences in molecular weight, polydispersity index (PDI), and defects in chain structure (*J. Am. Chem. Soc.* **2014**, 136, 11128-11133, and *Nano Energy.* **2024**, 123, 109397). The strategy of ternary copolymerization, by incorporating the third component to the host polymer chain, would inevitably exacerbate the disparities in molecular weight, PDI, and structural consistency of the polymers (*Adv. Funct. Mater.* **2024**, 34, 2315476). Unfortunately, the batch-to-batch variation of polymer donors would further strongly affect their aggregation behavior in solution and film formation, leading to poor repeatability of devices performance and mechanical robustness (*Energy Environ. Sci.* **2024**, 17, 3927-3936).

We revised the manuscript in the Introduction section in pp. 4-5: “**However, the random**

ternary copolymerization would exacerbate the polymer batch-to-batch variation with molecular weight difference or molecular structure defect and presents challenges for the repeatability of synthesis as well as the reliability of device photovoltaic performance and mechanical robustness^{39–43}.”.

We also added the corresponding references in References section on Page 32: “

39. Wan, Q. *et al.* Polymer acceptor with hydrogen-bonding functionality for efficient and mechanically robust ternary organic solar cells. *Chem. Mater.* **35**, 10476–10486 (2023).
40. Hendriks, K. H., Li, W., Heintges, H. L. & Janssen, A. J. Homocoupling defects in diketopyrrolopyrrole-based copolymers and their effect on photovoltaic performance. *J. Am. Chem. Soc.* **136**, 11128–11133 (2014).
41. He, Y., Huo, L. & Zheng, B. Advances of batch-variation control for photovoltaic polymers. *Nano Energy* **123**, 109397 (2024).
42. Deng, X. *et al.* High-performance terpolymers with well-defined structures facilitate pce over 19% for polymer solar cells. *Adv. Funct. Mater.* **34**, 2315476 (2024).
43. Zhang, T. *et al.* A highly crystalline donor enables over 17% efficiency for small-molecule organic solar cells. *Energy Environ. Sci.* **17**, 3927–3936 (2024).”.

(2) For the novelty of this work.

Some research groups had successfully introduced the H-bonding into the materials for obtaining high performance and mechanically robust organic semiconductors, however, these works are mainly focus on ternary copolymerization method (*Adv. Mater.* **2022**, 34, 2207544, *J. Am. Chem. Soc.* **2023**, 145, 11914–11920, and *Chem. Mater.* **2023**, 35, 10476-10486). As discussed above, the ternary copolymerization strategy would inevitably present challenges for the repeatability of synthesis, device photovoltaic performance and mechanical robustness. Moreover, the groups that can provide H-bonding are usually introduced by the third component in the ternary copolymerization, and the portion of the third unit is usually below 20%. Therefore, the portion of H-bonding introduced by this method is very limited.

Unlike the polymers, the small molecule acceptors (SMAs) possess a definite molecular structure, which allows them to have good repeatability and to more effectively introduce H-bonding into the active layer of OSCs. However, there was limited research on H-bonding modification for SMAs, which is mainly attributed to their strong π - π intermolecular interaction. The introduction of H-bonding into the SMAs will further strengthen intermolecular interaction,

which poses a huge challenge to coordinate the crystallinity of SMAs and corresponding blend film morphology. In this work, we **first attempted to introduce the dynamic chemical bonds (H-bonding) into the A-DA'D-A type SMAs** by selecting ethyl ester group as the unit providing proper H-bonding interactions, and **achieved both high photovoltaic performance and mechanical robustness OSCs**. During the research, by selecting an appropriate H-bonding unit and cleverly designing the distance between the unit and the conjugated core structure, **we solved the problem of difficulty in controlling the morphology of the blend films due to the enhanced crystallinity of the SMAs caused by introduction of H-bonding**. This has important implications for introducing dynamic chemical bonds into the design of high performance SMAs flexible photovoltaic materials that can be used in OSCs.

To emphasize the novelty and importance of this research, we revised the manuscript in the Introduction section on Page 5: “**Differing from the polymers, the SMAs possess a definite molecular structure and fixed molecular weight which allows them to have good repeatability and to more effectively introduce H-bonding into the active layer of OSCs. However, there was limited research on H-bonding modification for the SMAs, since they typically have the strong π - π intermolecular interaction and the introduction of H-bonding into the SMAs will further strengthen intermolecular interaction, which poses a huge challenge to coordinate the crystallinity of SMAs and corresponding blend film morphology.**”.

2. *Please explain how the COS increases from 4.3% to 7.0% are accompanied by a 50 MPa decrease in elastic moduli for PM6:BTA-C6 and PM6:BTA-E3, whereas the COS increases from 7.0% to 9.6% are accompanied by a 200 MPa decrease in elastic moduli for PM6:BTA-E3 and PM6:BTA-E6.*

Response: Thanks for the reviewer’s question. As far as we learned, the mechanical properties (such as COS, elastic moduli and toughness) of the blend film sample are its intrinsic properties and are independent to each other (*Chem. Rev.* **2017**, 117, 6467-6499). Besides, there is no apparent correlation among the mechanical property parameters in different samples. Each mechanical property parameter refers to different physical meaning, where the COS refers to the strain when fracture starts to appear, the elastic modulus indicates the ductility of the films and the toughness reflects the capability of films to resist crack and deformation (*Adv. Energy Mater.* **2022**, 12, 2201087).

According to these mechanical property parameters, we could gain insight into the inherent mechanical behavior of the films under the tensile process. As shown in Supplementary Fig. 49, the PM6:BTA-E3 blend film exhibited significantly enhanced COS value and toughness (COS value of 7.0% and toughness of 0.79 MJ m^{-3}) compared to the PM6:BTA-C6 counterpart (COS value of 4.3% and toughness of 0.29 MJ m^{-3}), while there was only a slight decrease in elastic modulus (404 MPa for PM6:BTA-E3 film and 453 MPa for PM6:BTA-C6 film). The results indicate that the PM6:BTA-E3 and PM6:BTA-C6 blend films possessed similar ductility, however, the **PM6:BTA-E3 blend film exhibited improved mechanical robustness capable of withstanding higher strain and stress due to the introduction of H-bonding**. Furthermore, the PM6:BTA-E6 blend film demonstrated the increased COS value of 9.6% and the significantly decreased elastic modulus of 215 MPa than the PM6:BTA-E3 blend film, which implies that as the ethyl ester side chain length elongating, **the PM6:BTA-E6 blend film exhibited the better strain resistance and ductility** than the PM6:BTA-E3 blend film.

3. *The authors attribute both the improved COS values and thermal stability to the introduction of hydrogen bonding. However, while PM6:BTA-E6 demonstrates the best COS value of 9.6%, PM6:BTA-E3 devices show the best thermal stability after introducing hydrogen bonding. Could you clarify the reason for this discrepancy?*

Response: Although the improved COS values and thermal stability of the blend films are both caused by the introduction of H-bonding into the SMAs, the factors affecting them are different.

Firstly, it is well known that the COS value primarily represents the resistance of the films against the strain, while the toughness most reflects the overall capability of the films to resist crack and deformation. The more effective H-bonding exist in the blended films, the better the mechanical stability of the films will be achieved. From BTA-E3 to BTA-E9, as their extended ethyl ester side chain positioned distantly from the central conjugated backbone, more H-bonding may be provided in the corresponding films owing to the significantly reduced steric effect. This change in structure and the number of effective H-bonding also leads to the improvement of molecular crystallization (see the *GIWAXS* results) and the increase of toughness (0.79 MJ m^{-3} , 0.83 MJ m^{-3} and 0.94 MJ m^{-3} for BTA-E3, BTA-E6 and BTA-E9, respectively). So the increase of machinal robustness of the film is closely related to the amount of effective H-bonding in the film.

In terms of thermal stability, besides the amount of effective H-bonding, the metastable

morphology of the blend films is also affected by the crystallization and miscibility of the donors and acceptors. In *GIWAXS* measurement, with the extension of the ethyl ester side chains in SMAs, the respective crystallization behavior of donor and acceptor components is enhanced, and the miscibility between the donor and acceptor is decreased. In addition, the χ values of BTA-E3, BTA-E6 and BTA-E9 increased from 0.10 *K* to 0.22 *K*, and to 0.39 *K*, which also indicates their decreased miscibility with PM6. It should be noted that when the blend film is exposed to light for a long time, the strong crystallinity and poor miscibility of the active layer materials will promote their self-aggregation, thus destroying the morphological stability of the film. This is the main reason for the decreased thermal stability of BTA-E6 and BTA-E9 than BTA-E3.

Overall, the morphology stability is affected by many factors. Therefore, we choose BTA-C6, which has the same alkyl chain length as BTA-E3, as the comparison material. The BTA-C6 and BTA-E3 have more similar molecular structure, so this comparison can more effectively and intuitively reflect the impact of H-bonding on the mechanical and thermal stability of the system. Thus, we also focus on the comparison of the differences between BTA-E3 and BTA-C6 in the manuscript.

Yongfang Li

Institute of Chemistry, Chinese Academy of Sciences

Beijing 100190, China.

REVIEWERS' COMMENTS

Reviewer #2 (Remarks to the Author):

All questions are suitably answered and the revised manuscript can be published as it is.

Reviewer #3 (Remarks to the Author):

The revised version provides substantial details and experiments to answer the reviewers' concerns, significantly improving the quality of the MS. I recommend publishing this paper in its current version. Congratulations to the authors.